# Query-Based Asymmetric Modeling with Decoupled Input-Output Rates for Speech Restoration

Ui-Hyeop Shin [1]   Jaehyun Ko [2]   Woocheol Jeong [2]   Hyung-Min Park [1,2]

## Abstract

Speech restoration aims to recover clean speech from degraded recordings affected by noise, reverberation, bandwidth reduction, or other distortions, where input and output sampling rates may differ. Existing approaches typically assume matched input–output rates and apply redundant resampling, limiting native multi-rate processing. We formulate this gap as the extended sampling-frequency-independent (xSFI) setting, where a model must operate under decoupled input–output rates, and propose TF-Restormer, a query-based xSFI modeling framework. The model encodes only the observed input band and synthesizes the unobserved high-frequency band through extension queries with band-partitioned cross-attention, yielding an asymmetric encoder–decoder that allocates capacity to analysis while keeping synthesis lightweight. Trained with a perceptual loss, a scaled log-spectral loss, and adversarial supervision via an SFI-STFT discriminator, TF-Restormer attains balanced fidelity–perceptual quality as a single unified model, without redundant resampling across denoising, dereverberation, bandwidth extension, and combined-distortion benchmarks under multiple sampling rates.

## 1. Introduction

Speech enhancement (Ephraim & Malah, 1984; Pascual et al., 2017) has historically progressed through isolated sub-tasks with dedicated models such as denoising (Hu et al., 2020; Ho et al., 2020), dereverberation (Han et al., 2015; Wang & Wang, 2020), declipping (Mack & Habets, 2019), and bandwidth extension or super-resolution (Liu et al., 2022a; Lee & Han, 2021). In real-world settings, multiple distortions often coincide and further obscure both magnitude and phase, making coherent and faithful speech restoration substantially more difficult. Crucially, this complexity is often accompanied by a fundamental mismatch between the native input bandwidth and the desired output sampling rate, a challenge that remains largely unaddressed by conventional task-specific frameworks.

This growing complexity has prompted a shift toward *general speech restoration* using generative models to handle diverse distortions (Liu et al., 2022b; Serrà et al., 2022). Vocoder-based approaches (Kumar et al., 2019; Kong et al., 2020a) reconstruct waveforms from compressed representations such as Mel features (Liu et al., 2022b; Babaev et al., 2024), improving perceptual quality but often treating speech as a semantic abstraction rather than a physical signal. Waveform- or complex-spectrum-based generative models (Oord et al., 2016; Serrà et al., 2022; Welker et al., 2022; Richter et al., 2023), including GAN- and diffusion-based methods, operate closer to the signal domain and can improve fidelity, but their temporal compression or iterative sampling limits streaming applicability.

Beyond the choice of generative modeling paradigm, existing approaches commonly assume *a fixed input–output sampling-rate* setting. Restoration and super-resolution models (Kim et al., 2024; Lu et al., 2025) typically resample inputs to a predetermined target rate, which fixes the spectral grid but ties the learned representation to a specific output resolution. Consequently, inputs with different native bandwidths require redundant resampling, and supporting multiple output rates often requires rate-specific models or repeated conversion. These limitations motivate a universal framework that analyzes the observed bandwidth directly and synthesizes arbitrary output rates without external resampling.

In parallel, discriminative models have been extensively developed in the complex STFT domain (Choi et al., 2018; Hu et al., 2020). More recently, time–frequency (TF) dual-path models have been introduced (Dang et al., 2022; Wang et al., 2023), which alternate sequence modeling along the time and frequency axes. By treating frequency bins as ordered sequences rather than static feature dimensions, this formu-

[1]Department of Electronic Engineering, Sogang University, Seoul, Korea [2]Department of Artificial Intelligence, Sogang University, Seoul, Korea. Correspondence to: Ui-Hyeop Shin <dmlguq123@sogang.ac.kr>, Hyung-Min Park <hpark@sogang.ac.kr>.

*Proceedings of the 43rd International Conference on Machine Learning*, Seoul, South Korea. PMLR 306, 2026. Copyright 2026 by the author(s).

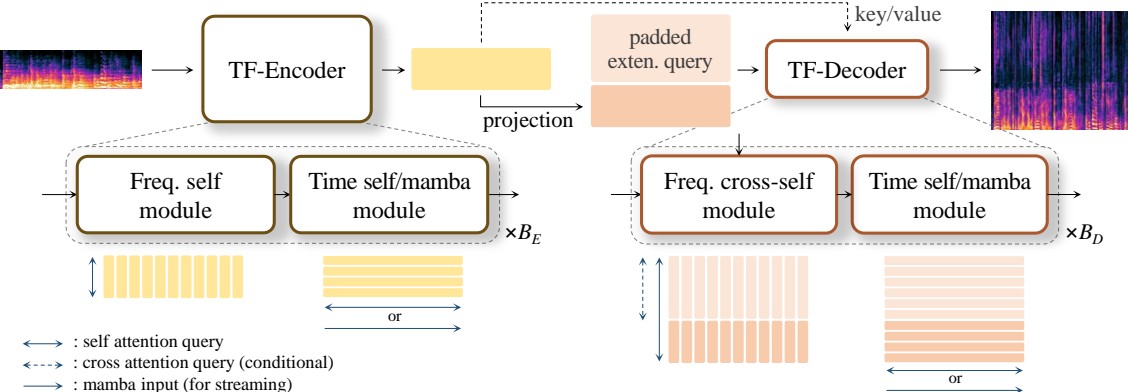

*Figure 1.* **Overall architecture of TF-Restormer.** A query-based asymmetric encoder–decoder analyzes the native input bandwidth and reconstructs missing high-frequency bands using learnable extension queries.

lation naturally supports sampling-frequency-independent (SFI) processing (Paulus & Torcoli, 2022; Zhang et al., 2023). The frequency axis can scale with the input sampling rate while preserving a consistent STFT frame duration. However, existing TF models still assume matched input–output rates. Moreover, because they preserve fine-grained spectral representations throughout the network, their computational cost grows substantially at higher sampling rates, limiting direct application to super-resolution.

These observations motivate *extended SFI* (xSFI) processing, which retains the strengths of TF dual-path SFI while operating under decoupled input–output rates rather than the matched-rate assumption. To this end, we propose *TF-Restormer*, an xSFI model based on an asymmetric encoder–decoder architecture for speech restoration under diverse degradations as shown in Figure 1. Inspired by masked autoencoders (MAE) (He et al., 2022), TF-Restormer concentrates heavy computation in a TF dual-path encoder that analyzes the observed input bandwidth, while a lightweight decoder reconstructs missing high-frequency components through learnable *extension queries* using a cross-self attention mechanism (Gupta et al., 2023). By explicitly separating input-band analysis from output-band synthesis, this asymmetric design enables a single model to operate across arbitrary input–output sampling rate pairs without external resampling, while preserving physical signal fidelity.

In summary, our contributions are as follows:

- We formulate speech restoration under *decoupled input–output sampling rates*, where the standard learning assumption of matched spectral resolutions no longer holds, and propose a *query-based asymmetric* TF encoder–decoder that analyzes the input bandwidth and synthesizes arbitrary output rates without external resampling.

- Our asymmetric design enables a *single model* to support general speech restoration including super-resolution, across heterogeneous input–output rate pairs, with unified training facilitated by a shared sampling-frequency-independent (SFI) STFT discriminator.

- We improve robustness and practicality by enhancing the frequency module with a projection-based *spectral inductive bias* and extending the time module to a causal variant, enabling stable restoration under severe degradations and real-time streaming inference.

- We propose a *scaled log-spectral loss* that stabilizes training under extreme distortions while mitigating over-smoothing, complementing perceptual (Babaev et al., 2024) and adversarial training (Mao et al., 2017).

## 2. Related Work

**Vocoder- and diffusion-based restoration** Generative approaches have significantly improved speech quality. Vocoder systems typically project speech into Mel features (Liu et al., 2022b; Babaev et al., 2024) or learned representations (Koizumi et al., 2023; Li et al., 2024) and synthesize waveforms with neural vocoders (Oord et al., 2016; Kumar et al., 2019; Kong et al., 2020a). They often achieve "studio-like" perceptual quality but reduce signal fidelity, since intermediate features serve as perceptual cues rather than physical signals. Diffusion-based methods from waveform (Kong et al., 2020b; Serrà et al., 2022; Scheibler et al., 2024) or STFT inputs (Lemercier et al., 2023; Richter et al., 2023) excel at generating diverse realizations. However, each approach often relies on temporal compression or iterative sampling, limiting real-time streaming. More critically, both paradigms are typically bound to a fixed-rate assumption, requiring external resampling to match a specific target bandwidth. In contrast, we propose a single-pass complex spectral prediction framework that avoids the abstraction bottleneck of vocoders and the complexity of diffusion, while supporting variable input–output rates.

**TF dual-path models** TF dual-path models (Dang et al., 2022) alternate sequence modeling along time and frequency, preserving spectral structure and naturally supporting sampling-frequency-independent (SFI) designs (Zhang et al., 2023). They have shown strong performance in en-

hancement (Cao et al., 2022; Lu et al., 2023; Chao et al., 2024) and separation (Saijo et al., 2024; Shin et al., 2025; Shin & Park, 2026). However, most prior models employ symmetric block designs for time and frequency processing, despite the distinct statistical characteristics of the two domains. Under challenging degradations, this limitation becomes more pronounced, motivating the use of projection-based frequency modules that inject spectral inductive bias for more robust high-frequency recovery. Furthermore, while existing dual-path models assume matched input-output rates, we extend this paradigm to decoupled rates as xSFI, realized through an asymmetric encoder-decoder that explicitly separates observed-band analysis from unobserved-band synthesis.

**Audio super-resolution** Conventional super-resolution models (Liu et al., 2022a; Han & Lee, 2022; Kim et al., 2024; Lu et al., 2025) typically assume a fixed target rate, upsampling inputs before processing to fill missing bands. While effective, this introduces redundancy and ties each model to a single output rate. Our framework moves beyond this fixed-rate constraint by framing super-resolution as a conditional spectral completion. Using *frequency extension queries* within an asymmetric encoder–decoder, TF-Restormer confines heavy processing to the input bandwidth and restores missing bands as needed, supporting arbitrary, user-specified output rates without rate-specific models.

## 3. TF-Restormer

### 3.1. xSFI input–output formulation

As an SFI model (Paulus & Torcoli, 2022), TF-Restormer addresses arbitrary input sampling rates $f_E$ by constructing STFT with a constant frame duration (SFI-STFT). Unlike conventional SFI that assumes matched input–output rates, we introduce xSFI formulation, enabling inference at user-specified output rates $f_D$. Given an input $x \in \mathbb{R}^{1 \times N_E}$ with sampling rate $f_E$, its STFT is $\mathbf{X} \in \mathbb{R}^{F_E \times T \times 2}$, where $F_E$ and $T$ are the number of frequency bins and frames. TF-Restormer then predicts $\mathbf{Y} \in \mathbb{R}^{F_D \times T \times 2}$ corresponding to an output $y \in \mathbb{R}^{1 \times N_D}$ at sampling rate $f_D$, satisfying $f_E : f_D = (F_E - 1) : (F_D - 1)$ under the assumption of consistent frame duration. To ensure universal applicability across sampling rates, we adopt a 40 ms window with a 20 ms hop. This choice yields integer sample counts, guaranteeing consistent STFT construction at typical rates of $\{8, 16, 22.05, 24, 32, 44.1, 48\}$ kHz without resampling.

### 3.2. Analysis encoder and extension decoder

As shown in Figure 1, TF-Restormer realizes the xSFI setting through a conditional information-flow structure between the encoder and the decoder: the encoder analyzes only the observed input band, the decoder synthesizes the missing band via extension queries with band-partitioned

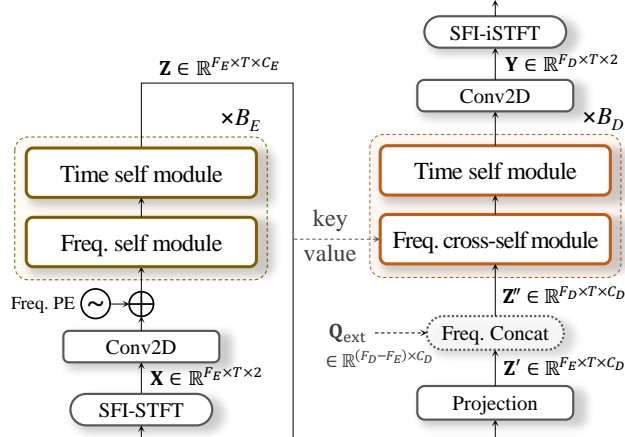

*Figure 2.* **Architecture of TF-Restormer.** The extension query $\mathbf{Q}_{\text{ext}}$ is concatenated to the projected feature $\mathbf{Z}'$ along the frequency axis, while encoder features $\mathbf{Z}$ provide the key and value for frequency cross-attention.

cross-attention, and when $f_E = f_D$ the cross-attention path is bypassed entirely. This rate-conditional asymmetry concentrates capacity on robust input analysis while keeping synthesis lightweight.

**TF-encoder for input analysis** Figure 2 details this realization at the module level. The TF-encoder analyzes the speech component from the input signals $\mathbf{X}$ within the native input bandwidth $F_E$, concentrating capacity on robust feature extraction. Before the TF-encoder, the input complex representation $\mathbf{X}$ is first projected to $C_E$ dimension by 2d convolution (Conv2D) layer with kernel size of (3,3), followed by layer normalization (LN) (Ba et al., 2016). Then, positional embeddings are added along the frequency axis. In the encoder, the projected feature is alternately processed by frequency and time self modules $B_E$ times to capture speech component.

**TF-decoder with extension query** The TF-decoder synthesizes the target output band by attending to the encoder features through band-partitioned cross-attention. The encoder features $\mathbf{Z}$ serve as key/value pairs in the frequency cross-self modules and, in parallel, are projected to the decoder channel dimension $C_D$ to form the lower-band decoder features $\mathbf{Z}'$. To reconstruct the missing high-frequency regions, we introduce learnable extension queries, $\mathbf{Q}_{\text{ext}} = [\mathbf{q}_{F_E+1}, \ldots, \mathbf{q}_{F_D}]^T \in \mathbb{R}^{(F_D - F_E) \times C_D}$, where $\mathbf{q}_f$ is a frequency-wise vector. These queries function as frequency-specific latent priors optimized to capture the characteristics of each sub-band. By slicing these queries from a unified master vector $\tilde{\mathbf{Q}}_{\text{ext}}$[1] defined over $[F_{\min}, F_{\max}]$, the decoder maintains consistent spectral alignment regardless of the

---

[1]The 40 ms SFI-STFT window fixes the frequency resolution at $\Delta_f = 25$ Hz, which divides every standard rate exactly into integer bin counts within $[F_{\min}, F_{\max}] = [161, 961]$ assuming the minimum input rates $f_E = 8$ kHz and maximum output rates $f_D = 48$ kHz.

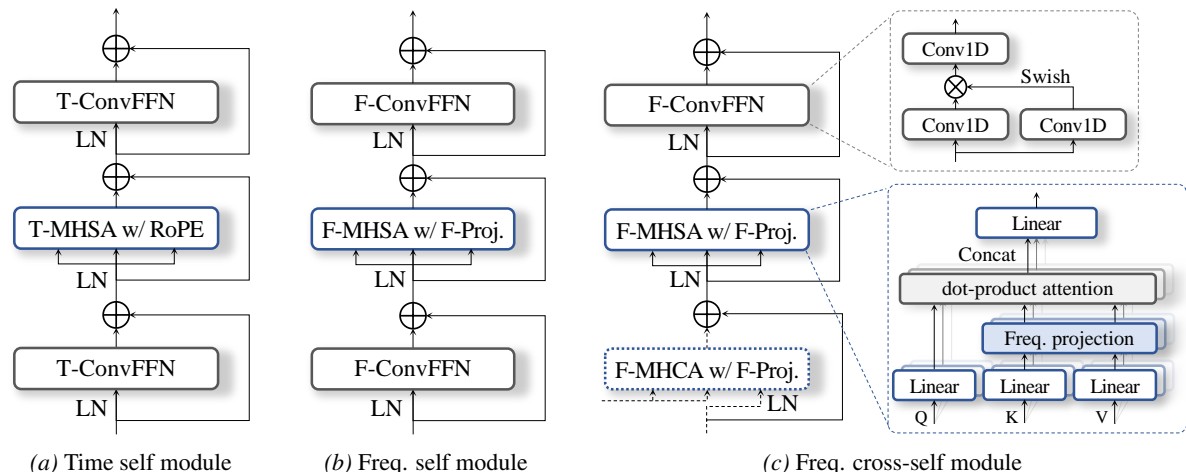

*(a)* Time self module      *(b)* Freq. self module      *(c)* Freq. cross-self module

*Figure 3.* Unit modules of the TF encoder and decoder. The self modules perform temporal or frequency-domain attention, while the frequency cross-self module uses encoder features as keys and values for cross-attention.

specific sampling rates. This ensures that the model treats a specific frequency bin with the same learned representation, facilitating rate-invariant generalization. In the frequency cross-self module, these queries explicitly guide the cross-attention mechanism to retrieve relevant acoustic context from the encoder features. Finally, the complex STFT values $\mathbf{Y}$ are estimated from the decoder features through a Conv2D layer.

### 3.3. TF dual-path module design

In TF dual-path modules, given the feature with shape $\mathbb{R}^{T \times F \times C}$ where $F \in \{F_E, F_D\}$, time modules process $F$ sequences with lengths of $T$ while frequency modules consider the feature as $T$ sequences with lengths of $F$ as illustrated in Figure 1.

**Time and frequency self module**    As shown in Figure 3a and 3b, self module consists of two macaron-style (Lu* et al., 2019) convolution feed-forward network (ConvFFN) with Conv1D with kernel size $K$ for capturing local contexts (Saijo et al., 2024; Shin & Park, 2026). In ConvFFN, the expansion factor is 3 with SwiGLU as hidden activation. Between ConvFFN modules, multi-head self-attention (MHSA) is used for global contexts with $H$ heads. The time module performs MHSA on temporal frames with rotary positional encoding (RoPE) (Su et al., 2024) to offer the relative positions. On the other hand, the frequency self module applies MHSA with the frequency projection layer to induce the structural bias as frequency bins are more static sequence with a fixed length, exhibiting a relatively consistent structural roles.

**Frequency cross-self module**    The frequency cross-self module in the decoder distinguishes between observed and unobserved frequency bins at the level of query construction. Specifically, for MHSA, the entire frequency axis is treated symmetrically, and all frequency bins participate as queries,

keys, and values. In contrast, MHCA is applied only to the unobserved high-frequency region $\mathbf{Q}_{\text{ext}}$ when $f_E < f_D$, while the key–value are derived from the encoder features $\mathbf{Z}$ corresponding to the observed input bandwidth. Consistent with the conditional information-flow structure, when $f_E = f_D$ the MHCA path is bypassed entirely.

**Attention with structural bias**    Linformer (Wang et al., 2020) introduced linear projections of key-value to reduce the computations, while MLP-Mixer (Tolstikhin et al., 2021) entirely replaced MHSA with static linear operations. Motivated by these insights (Quan & Li, 2024a), we incorporate a frequency linear projection (Figure 3c) to impose an inductive bias for the structural consistency of frequency bins on top of the benefits of dynamic attention. Since frequency bins exhibit consistent characteristics, we share the same projection layer across all modules and key-value mappings; this *shared* design emerges as the best choice in our ablation (Table 6b), consistent with the same pattern in Linformer and SpatialNet (Wang et al., 2020; Quan & Li, 2024a). Formally, for each head $h = 1, \ldots, H$, given key-value $\mathbf{K}_{h,c}, \mathbf{V}_{h,c} \in \mathbb{R}^{T \times F}$ at channel $c$, a projection matrix $\mathbf{A}_h \in \mathbb{R}^{F_{\max} \times F_{\text{proj}}}$ with dimension $F_{\text{proj}}$ and maximum bins $F_{\max}$ is applied as

$$\tilde{\mathbf{K}}_{h,c} = [\mathbf{K}_{h,c}, \mathbf{O}]\mathbf{A}_h \in \mathbb{R}^{T \times F_{\text{proj}}}, \quad 1 \le c \le C_h, \quad (1)$$

$$\tilde{\mathbf{V}}_{h,c} = [\mathbf{V}_{h,c}, \mathbf{O}]\mathbf{A}_h \in \mathbb{R}^{T \times F_{\text{proj}}}, \quad 1 \le c \le C_h, \quad (2)$$

where $\mathbf{O} \in \mathbb{R}^{T \times (F_{\max} - F)}$ is a zero-padding matrix.

**Streaming mode with Mamba**    The modular design of the TF dual-path model further enables a seamless extension to streaming mode by replacing the time module with Mamba blocks (Gu & Dao, 2024; Quan & Li, 2024b). Causal-masked attention is a viable alternative; we adopt Mamba following (Quan & Li, 2024b) for its constant-memory inference on long-form audio. Refer to Appendix C for detailed model configurations.

# 4. Training

The model is trained by two phases of pretraining and adversarial training. The model is trained with $f_D$ randomly selected from $\{16, 24, 44.1, 48\}$kHz at each step by downsampling target speech signals from VCTK dataset (Yamagishi et al., 2019). Based on speech sources, we simulated noisy reverberant signals by convolving the room impulse response (RIR) and noise samples from the DNS dataset (Reddy et al., 2020). We then applied various digital distortions including codecs and downsampled the signal to the sampling rates $f_E$ of 8k or 16kHz, which are common in practical restoration conditions (see Appendix A.1 for details). Because extension query could be undertrained if distribution of the input and output sample rates is unbalanced in training, we investigate these issues in Appendix F.

## 4.1. Pretraining

**Perceptual loss**   Following (Babaev et al., 2024), we incorporate a self-supervised learning (SSL)-based perceptual loss to stabilize adversarial training and encourage human-aligned quality. Specifically, extracting features from a pretrained SSL model for both the enhanced and clean waveforms, we minimize the mean-squared-error between these representations:

$$\mathcal{L}_{\mathrm{p}}(\theta) = \mathbb{E}_{m,n}\big[\,|\phi(g_\theta(x))_{m,n} - \phi(s)_{m,n}|^2\big], \quad (3)$$

given that $g_\theta(x)$ is output of restoration model $g_\theta(\cdot)$ with parameters of $\theta$. $\phi(\cdot)_{m,n}$ denotes the $m$-th element of $n$-th frame from its feature map. We utilize WavLM-conv (Chen et al., 2022b) as in the previous study (Babaev et al., 2024).

**Proposed scaled log-spectral loss**   Because the perceptual loss is restricted to 16 kHz and it is beneficial to guide spectral details to complement the looseness of perceptual loss, a previous work adopted an $\ell_1$ distance on the magnitude spectrum (Babaev et al., 2024). However, because TF-Restormer operates directly on the complex spectrum, the model can be explicitly supervised on both real and imaginary components in addition to the magnitude. Therefore, when denoting the STFT of target signal $s$ by $S_{m,tf} = |S_{r,tf} + jS_{i,tf}|$ and that of model's predicted signal $g_\theta(x)$ by $Y_{m,tf} = |Y_{r,tf} + jY_{i,tf}|$, we can extend to the complex domain as $\mathcal{L}_{\ell_1}(\theta) = \sum_{c\in\mathcal{C}} \alpha_c \cdot \mathbb{E}_{t,f}[|Y_{c,tf} - S_{c,tf}|]$, where $\mathcal{C} = \{r, i, m\}$ denotes the component index set and $\alpha_c$ are component weights.

However, even with complex supervision, severely degraded or missing high-frequency regions yield unstable gradients and drive the model toward oversmoothing under $\ell_1/\ell_2$ supervision (Babaev et al., 2024). To address this, we design the loss to satisfy three properties: (i) a *monotonically decreasing gradient* at large errors, (ii) a *unit-height gradient peak* matching the $\ell_1$ upper bound, which keeps the learning signal uniform, and (iii) $w_{tf}$ acting as a *transition-point* pa-

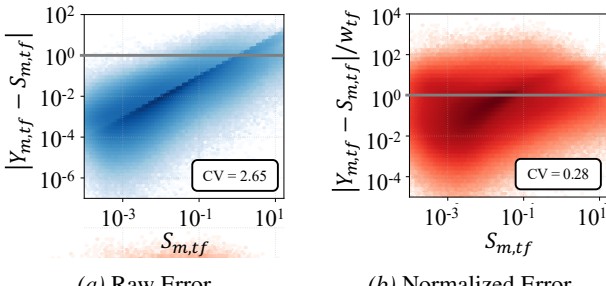

*(a)* Raw Error          *(b)* Normalized Error

*Figure 4.* Spectral error versus source magnitude $S_{m,tf}$ on VCTK-SR(noisy-distorted, $16 \to 48$ kHz) testset from Table 3. The gray horizontal line marks the log1p transition point at unit error (gradient is $\approx 1$ below and saturates toward 0 above). Insets report the coefficient of variation (CV) of the plotted error: (a) raw error and (b) normalized error with $w_{tf} = \mathrm{E}_t[S_{m,tf}]$.

rameter for the saturation onset. Among functions satisfying all three[2], the scaled log1p form $w\log(1 + |d|/w)$ provides the simplest single-parameter form:

$$\mathcal{L}_{\mathrm{s}}(\theta) = \sum_{c\in\mathcal{C}} \alpha_c \cdot \mathbb{E}_{t,f}\left[\,w_{tf}\log\left(1 + \frac{|Y_{c,tf} - S_{c,tf}|}{w_{tf}}\right)\right], \quad (4)$$

where $w_{tf}$ is scale factor that controls the relative scaling of gradient.

For choosing the weight value $w_{tf}$, we empirically observed that the distance $|Y_{c,tf} - S_{c,tf}|$ is heteroscedastic and proportional to the source magnitude $S_{m,tf}$ (Figure 4a): low-magnitude bins lie far below the unit transition line, where log1p reduces to a near-$\ell_1$ gradient and still introduces the oversmoothing, while high-magnitude bins drift above the line into the saturated regime. Setting $w_{tf} = \mathrm{E}_t[S_{m,tf}]$ by averaging over the frames of the ground-truth target aligns the error distribution into a magnitude-invariant band centered around the transition line, so every bin lives inside the log1p regime as intended: the gradient concentrates on the well-aligned regions for refinement while large-distance outliers are descended toward zero, suppressing oversmoothing. We use $\alpha_m = 0.6$, $\alpha_r = 0.2$, and $\alpha_i = 0.2$.

## 4.2. Adversarial training

After pretraining the generator with $\mathcal{L}_{\mathrm{pre}}$, we introduce an adversarial loss component to reduce the artifacts and predict severely distorted or missing components. For adversarial training, we attach multi-scale STFT discriminators (Défossez et al., 2023) as $i$-th discriminator of $\varphi_i$ and apply least square GAN (LS-GAN) loss (Mao et al., 2017). For generator, generator LS-GAN and feature-matching loss (Kumar et al., 2019) terms are added, respectively:

---

[2]Appendix H extends the comparison to the broader robust-loss family and leaves open whether bilateral-saturation alternatives better suit spectral regression.

$$\mathcal{L}_{\text{gen}}(\theta) = \lambda_{\text{g}}\mathcal{L}_{\text{g}}(\theta) + \lambda_{\text{fm}}\mathcal{L}_{\text{fm}}(\theta)$$
$$+ \lambda_{\text{p}}\mathcal{L}_{\text{p}}(\theta) + \lambda_{\text{s}}\mathcal{L}_{\text{s}}(\theta) + \lambda_{\text{hf}}\mathcal{L}_{\text{hf}}(\theta), \quad (5)$$
$$\mathcal{L}_{\text{disc}}(\varphi_i) = \mathcal{L}_{\text{d}}(\varphi_i), \quad i = 1, ..., I. \quad (6)$$

where $\mathcal{L}_{\text{hf}} = \mathcal{L}_{\text{pesq}} + 10 \cdot \mathcal{L}_{\text{utmos}}$ is additional human-feedback loss (Babaev et al., 2024) for aesthetic quality with differentiable PESQ loss and UTMOS loss (Saeki et al., 2022). We performed adversarial training using $\mathcal{L}_{\text{gen}}(\theta)$ with $\lambda_{\text{g}} = 0.005$, $\lambda_{\text{fm}} = 0.1$, $\lambda_{\text{p}} = 100$, $\lambda_{\text{s}} = 1$, and $\lambda_{\text{hf}} = 0.0001$. Notably, we assign small weights to $\mathcal{L}_{\text{g}}$ and $\mathcal{L}_{\text{hf}}$ to avoid excessive generation artifacts.

**Proposed multi-scale SFI-STFT discriminators** In conventional adversarial training, a dedicated generator for each target sampling rate is trained with a corresponding discriminator as well (Défossez et al., 2023; Babaev et al., 2024; Ju et al., 2024), which introduces implementation overhead depending on the output sample rates. For training of a single generator across diverse rates, we adopt the strided-Conv2D STFT discriminator (Défossez et al., 2023) producing two-dimensional maps for local real/fake supervision in the time-frequency plane, and replace its STFT front-end with the SFI-STFT formulation so that a single discriminator operates across sampling rates with a consistent physical frame duration. This design maintains sensitivity to spectral structure while remaining agnostic to absolute frequency resolution, thereby supporting adversarial training across different rates without redundant resampling or rate-specific discriminators. We employ 5 SFI discriminators with STFT window durations of $\{20, 40, 60, 80, 100\}$ ms.

## 5. Evaluation

### 5.1. Test dataset and metrics

We evaluate a unified TF-Restormer on various datasets to validate robustness across heterogeneous distortion conditions and arbitrary input–output sample rates. For denoising (DN) and speech super-resolution (SSR), we additionally report dedicated versions of TF-Restormer for fair comparison; their training configurations are summarized in Appendix A.3.

**UNIVERSE data for general speech restoration (GSR)** As GSR model, we evaluate on 100 synthetic samples generated by UNIVERSE authors (Serrà et al., 2022) to ensure comparability to prior works in $f_E = f_D = 16$kHz setting. The dataset introduces diverse simulated degradations such as bandpass filtering, reverberation, codec compression, and transmission artifacts.

**VCTK+DEMAND for DN** We additionally evaluated the well-known Valentini denoising dataset (Valentini-Botinhao & others, 2017) for direct comparison with conventional enhancement models as speech enhancement benchmarking. The evaluation set (824 utterances) consists of noisy mix-

*Table 1.* Results on UNIVERSE data for GSR. [†]We utilized pretrained models from implementation code from UNIVERSE++ (Scheibler et al., 2024). [‡]The results are reported in the original paper (Babaev et al., 2024)

| Model | Signal fidelity | | | | | Perceptual quality | |
|---|---|---|---|---|---|---|---|
| | PESQ$^\uparrow$ | SDR$^\uparrow$ | LSD$^\downarrow$ | MCD$^\downarrow$ | sBERT$^\uparrow$ | UTMOS$^\uparrow$ | DNSMOS$^\uparrow$ |
| Input | 1.55 | 5.58 | 1.89 | 10.21 | 0.84 | 2.19 | 2.23 |
| Ground Truth | 4.50 | $\infty$ | 0.00 | 0.00 | 1.00 | 4.26 | 3.33 |
| VoiceFixer | 1.77 | -5.68 | 1.49 | 10.50 | 0.84 | 2.83 | 2.99 |
| StoRM | 1.76 | 9.01 | 1.67 | 6.87 | 0.84 | 2.70 | 2.94 |
| UNIVERSE$^\dagger$ | 1.74 | 7.73 | 1.92 | 6.25 | 0.79 | 2.64 | 2.73 |
| UNIVERSE++$^\dagger$ | 1.80 | 8.42 | 1.76 | 5.96 | 0.81 | 2.71 | 2.82 |
| TF-Locoformer | 2.13 | **11.61** | 2.00 | 6.26 | 0.89 | 2.95 | 2.86 |
| FINALLY | - | - | - | - | - | 4.21$^\ddagger$ | 3.25$^\ddagger$ |
| TF-Restormer (*off*) | **2.30** | 11.12 | **1.45** | **5.08** | **0.91** | 4.08 | **3.25** |
| TF-Restormer (*on*) | 2.00 | 8.89 | 1.47 | 6.01 | 0.87 | 3.77 | 3.14 |

tures from two speakers under four SNR conditions (17.5, 12.5, 7.5, and 2.5 dB).

**VCTK for SSR** For SSR evaluation, we construct paired data by downsampling 48 kHz clean utterances from the VCTK-0.92 dataset (Yamagishi et al., 2019). Beyond the clean case, we also create noisy-distorted conditions by adding degradations such as noise, reverberation, band-pass filtering, and codec effects, enabling a comprehensive evaluation of GSR with SSR. Note that the training simulation follows a similar procedure, which may provide a slight advantage to our model.

For the evaluation, we adopt non-intrusive perceptual estimators for mean opinion score (MOS): DNSMOS (Reddy et al., 2022), UTMOS (Saeki et al., 2022), and NISQA (Mittag et al., 2021) to assess the perceptual quality. Also, to assess the perceptual signal fidelity of restored signal compared to the reference, we employ perceptual evaluation of speech quality (PESQ) (Rix et al., 2001), signal-to-distortion ratio (SDR) (Le Roux et al., 2019), log-spectral distance (LSD), mel-cepstral distortion (MCD) (Fukada et al., 1992). In addition, to evaluate the reference-aware speech generation quality by capturing semantic congruence, we report SpeechBERTScore(sBERT) (Saeki et al., 2024).

### 5.2. Validation of stability across diverse scenarios

In this section, we validate the operational stability of TF-Restormer across a wide spectrum of restoration scenarios. For the UNIVERSE dataset, we consider Voice-Fixer (Liu et al., 2022b) as a Mel vocoder-based baseline, StoRM (Lemercier et al., 2023), UNIVERSE (Serrà et al., 2022), and UNIVERSE++ (Scheibler et al., 2024) as diffusion-based baselines, TF-Locoformer as a recent TF dual-path Transformer model, and FINALLY (Babaev et al., 2024) as a latest strong Mel-vocoder method. As shown in Table 1, VoiceFixer improves MOS but sacrifices fidelity due to its Mel representation, while FINALLY achieves the highest perceptual quality yet lacks signal fidelity, a

*Table 2.* Results on VCTK+DEMAND for denoising task. [†]Dedicated models trained specifically for denoising.

| Model | Signal fidelity | | | | | Perceptual quality | |
|---|---|---|---|---|---|---|---|
| | PESQ[↑] | SDR[↑] | LSD[↓] | MCD[↓] | sBERT[↑] | UTMOS[↑] | DNSMOS[↑] |
| Input | 1.98 | 8.56 | 1.27 | 5.40 | 0.91 | 2.90 | 2.45 |
| Ground Truth | 4.50 | ∞ | 0.00 | 0.00 | 1.00 | 4.07 | 3.16 |
| DB-AIAT | 3.27 | 21.30 | 0.90 | 1.77 | 0.95 | 3.83 | 3.13 |
| MP-SENet | 3.61 | 21.03 | 0.85 | 1.58 | 0.95 | 3.86 | 3.12 |
| TF-Locoformer | 3.30 | **23.82** | 0.92 | 3.58 | 0.95 | 3.93 | 3.20 |
| VoiceFixer | 2.40 | -1.12 | 0.97 | 7.40 | 0.90 | 3.50 | 3.08 |
| UNIVERSE | 2.84 | 18.77 | 1.17 | 2.20 | 0.92 | 3.75 | 3.03 |
| FINALLY | 2.94 | 4.60 | - | - | - | **4.32** | **3.22** |
| TF-Restormer (*off*) | 3.41 | 19.45 | 0.75 | 1.54 | **0.95** | 4.14 | 3.14 |
| TF-Restormer (*off*)[†] | **3.63** | 22.81 | **0.73** | **1.49** | 0.95 | 4.04 | 3.13 |
| TF-Restormer (*on*) | 2.89 | 16.43 | 0.85 | 2.16 | 0.93 | 4.05 | 3.09 |

*Table 3.* Results on VCTK for SSR under clean and noisy-distorted conditions. [†]The models require fixed output sampling rates $f'$, thus evaluated by upsampling the input of $f_E$ to $f' \geq f_D$ and downsampling the output back to the target rate $f_D$. [‡]Dedicated models trained specifically for super-resolution.

| Method | $8 \to 16$kHz | | $8 \to 24$kHz | | $8 \to 44.1$kHz | | $16 \to 48$kHz | |
|---|---|---|---|---|---|---|---|---|
| | LSD[↓] | NISQA[↑] | LSD[↓] | NISQA[↑] | LSD[↓] | NISQA[↑] | LSD[↓] | NISQA[↑] |
| *clean (SSR only)* | | | | | | | | |
| Input | 2.53 | 3.78 | 2.91 | 3.78 | 3.44 | 3.78 | 3.17 | 4.40 |
| NVSR[†] | 0.83 | 4.15 | 0.89 | 4.24 | 0.94 | 4.16 | - | - |
| Frepainter[†] | 1.33 | 3.94 | 1.40 | 3.79 | 1.37 | 3.71 | 1.31 | 4.01 |
| AP-BWE[†] | 0.90 | 4.20 | 0.86 | 4.34 | 0.88 | 4.26 | 0.85 | 4.33 |
| VoiceFixer[†] | 1.05 | 4.20 | 1.05 | 4.27 | 1.06 | 4.21 | - | - |
| TF-Restormer (*off*) | 0.89 | **4.53** | 0.95 | **4.61** | 1.01 | **4.54** | 0.97 | **4.62** |
| TF-Restormer (*off*)[‡] | **0.81** | 4.42 | **0.82** | 4.58 | **0.82** | 4.40 | **0.81** | 4.57 |
| *Noisy-distorted (GSR + SSR)* | | | | | | | | |
| Input | 3.36 | 1.91 | 3.49 | 1.91 | 3.64 | 1.91 | 3.48 | 1.73 |
| VoiceFixer[†] | 1.36 | 3.73 | 1.35 | 3.91 | 1.40 | 3.80 | - | - |
| StoRM | 1.76 | 3.97 | - | - | - | - | - | - |
| UNIVERSE++ | 1.79 | 3.39 | - | - | - | - | - | - |
| TF-Restormer (*off*) | **1.16** | **4.49** | **1.21** | **4.54** | **1.18** | **4.52** | **1.18** | **4.54** |
| TF-Restormer (*on*) | 1.30 | 4.42 | 1.31 | 4.49 | 1.30 | 4.46 | 1.26 | 4.46 |

trend confirmed in Table 2. Diffusion-based methods yield more balanced results by directly operating in the waveform or complex STFT. TF-Locoformer preserves signal-level fidelity but suffers from residual perceptual artifacts and failure to recover lost details and naturalness (MOS, LSD). In contrast, TF-Restormer provides balanced improvements for signal fidelity and perceptual quality, with its streaming variant maintaining competitive effectiveness under causal constraints. This indicates its robustness across diverse degradations in a universal restoration setting. Note that all the compared models are offline methods.

Next, we evaluate TF-Restormer on the VCTK+DEMAND focusing on denoising. In Table 2, we compare against DB-AIAT (Yu et al., 2022), MP-SENet (Lu et al., 2023), and TF-Locoformer as dedicated denoising models, and Voice-Fixer, UNIVERSE, and FINALLY as universal restoration baselines. Since the input speech is already well preserved and only corrupted by additive noise, it favors models that minimize unnecessary generation and faithfully retain the input signal. Accordingly, dedicated denoising models outperform universal restoration models in terms of signal fidelity, as they are optimized to suppress noise without altering intact regions. In contrast, restoration models risk degrading reliability by over-modifying clean inputs, making them less trustworthy for such simple cases. While not surpassing dedicated denoising models in raw signal metrics, TF-Restormer achieves more consistent gains, showing strong generalization despite being designed for universal restoration. We additionally include a dedicated TF-Restormer variant; as expected, this task-matched version achieves the higher signal-fidelity scores.

Finally, we experiment on the SSR task in Table 3, using a single model that directly supports arbitrary output sampling rates. For clean cases, we compare against dedicated super-resolution models: NVSR (Liu et al., 2022a), Frepainter (Kim et al., 2024), and AP-BWE (Lu et al., 2025), as well as VoiceFixer as a universal restoration baseline. As in Table 2, since the low-band of the input speech remains in-

tact, dedicated models that concentrate on reconstructing the upper bands are favored. Unlike conventional approaches that rely on fixed input–output rates and often require zero-padding or redundant resampling, TF-Restormer leverages extension queries to dynamically expand the spectrum. With this versatility, TF-Restormer shows stable performance comparable to the dedicated models, faithfully retaining clean low-frequency regions while effectively generating high-frequency components. When the training is optimally aligned with the conventional method, the dedicated version of the proposed model shows improved results. In addition, under noisy-distorted conditions, TF-Restormer simultaneously restores corrupted regions and reconstructs missing high bands, demonstrating robust generalization beyond pure super-resolution. Overall, these results suggest the advantage of our model as a universal restoration framework that achieves bandwidth extension without sacrificing signal fidelity or requiring explicit resampling.

### 5.3. MOS evaluation on additional datasets

In Table 4, we evaluate TF-Restormer on real-recorded Vox-Celeb (Nagrani et al., 2017) utterances, the URGENT 2025 blind test set (Saijo et al., 2025), the DNS Challenge 2020 real recordings (Reddy et al., 2020), and the REVERB Challenge (Kinoshita et al., 2013) real recordings to examine robustness in real and out-of-distribution(OOD) conditions. As paired references are unavailable for these datasets, we report non-intrusive MOS predictors here and we refer to audio demos to complement the lack of MOS evaluation.

**VoxCeleb real recordings** We evaluate TF-Restormer on 50 real-recorded utterances (Su et al., 2020) from Vox-Celeb1 and compare with conventional methods including DEMUCS (Défossez et al., 2019) and HiFi-GAN-2 (Su

*Table 4.* Evaluation of non-intrusive MOS results on real-recorded data from VoxCeleb, URGENT 2025 blind, DNS 2020, and RE-VERB.

| Model | UTMOS | DNSMOS |
|---|---|---|
| Input | 2.76 | 2.72 |
| VoiceFixer | 2.60 | 3.08 |
| DEMUCS | 3.51 | 3.27 |
| StoRM | 3.29 | 3.17 |
| HiFi-GAN-2 | 3.67 | 3.32 |
| FINALLY | **4.05** | 3.31 |
| TF-Restormer | 3.98 | **3.34** |

*(a) VoxCeleb*

| Model | UTMOS | NISQA | DNSMOS |
|---|---|---|---|
| Input | 1.55 | 1.58 | 1.90 |
| Bobbsun(R.1) | 2.09 | 3.22 | 2.88 |
| rc(R.2) | 2.03 | 2.92 | 2.83 |
| Xiaobin(R.3) | 2.16 | 3.24 | 2.92 |
| wataru9871(R.13) | 2.53 | 3.74 | 3.10 |
| LLaSE-G1 | 2.09 | 2.93 | 2.80 |
| UniSE | 2.85 | 3.72 | **3.17** |
| TF-Restormer | **3.37** | **4.37** | 3.13 |

*(b) URGENT 2025 blind*

| Model | UTMOS | NISQA | DNSMOS |
|---|---|---|---|
| Input | 1.94 | 2.16 | 2.21 |
| Miipher | **3.91** | 4.12 | **3.17** |
| VoiceFixer | 2.35 | 3.53 | 2.88 |
| Resem.-Enh. | 2.77 | 4.33 | 3.10 |
| UNIVERSE++ | 2.31 | 3.32 | 2.64 |
| AnyEnhance | — | — | 3.17 |
| TF-Restormer | 3.50 | **4.36** | 3.13 |

*(c) DNS 2020*

| Model | UTMOS | NISQA | DNSMOS |
|---|---|---|---|
| Input | 1.56 | 1.72 | 1.35 |
| TF-GridNet | 2.34 | 1.95 | 2.93 |
| MP-SENet | 2.09 | 1.84 | 3.01 |
| TF-Locoformer | 2.13 | 1.64 | 2.85 |
| TF-Restormer | **3.14** | **2.13** | **3.30** |

*(d) REVERB*

*Table 5.* Ablation study on encoder-decoder design. VCTK-SR(noisy-distorted) denotes noisy-distorted input from VCTK data in Table 3.

| Case | Size(M) | MAC(G) | LSD↓ | NISQA↑ |
|---|---|---|---|---|
| **VCTK** (*SSR, 8 → 16*kHz) | | | | |
| encoder-only | 11.6 | 151.3 | 2.12 | 4.21 |
| encoder-decoder w/o MHCA | 30.8 | 252.4 | 1.04 | 4.48 |
| encoder-decoder w/ MHCA | 30.1 | 240.8 | **0.89** | **4.53** |
| encoder-decoder w/ MHCA(*small*) | 10.9 | 89.2 | 1.36 | 4.38 |
| **VCTK** (*SSR, 8 → 44.1*kHz) | | | | |
| encoder-only | 11.6 | 415.1 | 3.25 | 3.49 |
| encoder-decoder w/o MHCA | 30.8 | 340.4 | 1.35 | 4.26 |
| encoder-decoder w/ MHCA | 30.1 | 308.4 | **1.01** | **4.54** |
| encoder-decoder w/ MHCA(*small*) | 10.9 | 156.8 | 1.44 | 4.21 |
| **VCTK-SR(noisy-distorted)** (*SSR+GSR, 8 → 16*kHz) | | | | |
| encoder-only | 11.6 | 151.3 | 2.23 | 3.72 |
| encoder-decoder w/o MHCA | 30.8 | 252.4 | 1.20 | 4.33 |
| encoder-decoder w/ MHCA | 30.1 | 240.8 | **1.16** | **4.54** |
| encoder-decoder w/ MHCA(*small*) | 10.9 | 89.2 | 1.48 | 4.20 |

ble 4d, TF-Restormer attains the strongest results across all baselines, indicating natural-sounding dereverberation despite never having seen REVERB-specific training data.

### 5.4. Ablation study

To validate the effects of the proposed methods, we conduct an ablation study on scaled log-spectral loss, decoder design, and frequency projection module.

**Encoder-decoder design** In Table 5, we first analyze the contribution of the encoder–decoder structure (See Appendix F for detailed illustration.). The *Encoder-only* model removes the decoder entirely and applies nine encoder blocks after padding the extension queries at the input. Although this variant has a small parameter count (11.6M), its MACs are extremely large (151-415G), since the encoder must jointly infer the observed low band and synthesize the missing high-frequency components. The *w/o MHCA* variant restores the encoder–decoder structure but replaces the freq cross-self module with the freq. self module used in the encoder, preventing the decoder from conditioning on encoder features. Our proposed *w/ MHCA* model incorporates cross–self frequency attention, enabling more reliable reconstruction through encoder-conditioned queries.

To further isolate the effect of architectural design from model size, we additionally include a reduced version of the proposed model (w/ MHCA(*small*)), whose parameter count matches that of the encoder-only configuration. Despite having far fewer MACs than encoder-only, this size-matched variant consistently outperforms the encoder-only model across all bandwidth settings (Table 5(b)), confirming that the gains arise from the explicit separation of analysis and reconstruction and the use of cross-attention rather than increased parameter count or computational cost.

**SFI-STFT discriminator** As shown in Table 6a, *a shared SFI*-STFT discriminator consistently outperforms *separate*

---

et al., 2021). As shown in Table 4a, TF-Restormer achieves perceptual MOS scores (UTMOS, DNSMOS) comparable to recent vocoder- and diffusion-based models. While FI-NALLY (Babaev et al., 2024) remains one of the strongest perceptual-quality systems, our unified architecture delivers similarly natural outputs, demonstrating competitive robustness on real speech recordings.

**URGENT 2025 blind testsets** Finally, we report non-intrusive MOS metrics on the URGENT 2025 blind test set compared to participating teams and latest models including LLaSE-G1 (Kang et al., 2025) and UniSE (Yan et al., 2025). We report this as OOD transfer evidence rather than a controlled in-domain benchmark—our model is trained exclusively on VCTK without URGENT-specific data—and TF-Restormer produces natural-sounding outputs with strong non-intrusive MOS, indicating stable transfer to URGENT blind conditions.

**DNS Challenge 2020 real recordings** We further evaluate TF-Restormer on the DNS Challenge 2020 real-recording subset as an OOD restoration model (no DNS-specific re-training). As shown in Table 4c, TF-Restormer attains the strongest NISQA among the unified-restoration baselines (Miipher (Koizumi et al., 2023), VoiceFixer, Resemble-Enhance[3], UNIVERSE++, AnyEnhance (Zhang et al., 2025)), with UTMOS and DNSMOS competitive with the best of these.

**REVERB Challenge real recordings** We additionally evaluate on the REVERB Challenge real-recording subset as an OOD restoration model (no REVERB-specific re-training), comparing against discriminative dereverberation baselines (TF-GridNet, MP-SENet, TF-Locoformer) trained on REVERB challenge training dataset. As shown in Ta-

---

[3] https://github.com/resemble-ai/resemble-enhance

*Table 6.* Ablation study on SFI-discriminator and frequency projection layer in freq. module

| STFT-Disc. | PESQ | UTMOS |
|---|---|---|
| **UNIVERSE (*GSR*)** | | |
| Separate | 2.27 | 4.08 |
| Shared (SFI) | **2.29** | **4.10** |
| **VCTK+DEMAND (*DN*)** | | |
| Separate | 3.28 | 4.07 |
| Shared (SFI) | **3.41** | **4.14** |
| **VCTK (*SSR*, 8 → *16*kHz)** | | |
| Separate | 3.64 | 4.10 |
| Shared (SFI) | **3.70** | **4.11** |

*(a)* Effect of discriminator

| Case | Size(M) | PESQ | UTMOS |
|---|---|---|---|
| **UNIVERSE (*GSR*)** | | | |
| w/o F-proj. | 28.1 | 2.26 | 3.90 |
| w/ F-proj.(*sep.*) | 63.6 | **2.31** | **4.11** |
| w/ F-proj.(*sha.*) | 30.1 | 2.29 | 4.10 |
| **VCTK+DEMAND (*DN*)** | | | |
| w/o F-proj. | 28.1 | 3.21 | 4.03 |
| w/ F-proj.(*sep.*) | 63.6 | 3.38 | **4.15** |
| w/ F-proj.(*sha.*) | 30.1 | **3.41** | 4.14 |
| **VCTK (*SSR*, 8 → *16*kHz)** | | | |
| w/o F-proj. | 28.1 | 3.54 | 3.97 |
| w/ F-proj.(*sep.*) | 63.6 | 3.55 | 3.94 |
| w/ F-proj.(*sha.*) | 30.1 | **3.70** | **4.11** |

*(b)* Effects of frequency projection

*Table 7.* Ablation study on spectral loss. log1p denotes $\log(1 + d)$ while s-log1p is the proposed scaled log1p $w \log(1 + d/w)$ where $d$ is $\ell_1$ distance.

| Type of spectral loss | $\mathcal{L}_\mathrm{p}$ | UNIVERSE(*GSR*) | | | VCTK(*SSR,8→16*kHz) | | |
|---|---|---|---|---|---|---|---|
| | | PESQ$^\uparrow$ | MCD$^\downarrow$ | UTMOS$^\uparrow$ | PESQ$^\uparrow$ | MCD$^\downarrow$ | UTMOS$^\uparrow$ |
| None | ✓ | 1.85 | 7.23 | 4.02 | 3.05 | 2.47 | 4.11 |
| $\ell_1$-norm (mag.) | ✓ | 2.07 | 6.03 | **3.82** | 3.42 | 2.10 | **4.10** |
| $\ell_1$-norm | ✓ | **2.23** | **5.70** | 3.76 | **3.48** | **1.86** | 4.07 |
| $\ell_2$-norm | ✓ | 2.21 | 5.81 | 3.70 | 3.44 | 1.88 | 4.06 |
| $\ell_1$-norm | | 2.19 | 5.89 | 3.71 | 3.35 | 2.23 | 3.91 |
| log1p $(w = 1)$ | ✓ | 2.25 | 5.72 | 3.79 | 3.53 | 1.83 | 4.06 |
| s-log1p $(w = 10^{-3})$ | ✓ | **2.27** | **5.17** | **3.98** | **3.67** | **1.37** | 4.07 |
| s-log1p $(w = 10^{-4})$ | ✓ | 2.01 | 5.94 | 4.07 | 3.40 | 2.43 | **4.10** |
| s-log1p $(w = 10^{-3})$ | | 2.18 | 6.05 | 3.74 | 3.27 | 1.88 | 3.94 |
| s-log1p (adap. $w_{tf}$) | ✓ | **2.29** | **4.96** | **4.10** | **3.70** | **1.29** | **4.10** |

*rate-specific* discriminators across all tasks. Because the SFI representation aligns TF structure across sampling rates, the unified discriminator receives more coherent supervision and produces more stable gradients. In contrast, separate discriminators see only partial bandwidth conditions, leading to weaker adversarial signals. These results confirm that the shared SFI design is more effective for multi-rate restoration.

**Effects of frequency projection**  Finally, we examine the influence of the *frequency-projection (F-proj.)* module. Introducing projection provides an explicit structural prior along the frequency axis, which stabilizes training and yields consistent improvements over the no-projection baseline across tasks. In the shared setting, a single projection matrix $\mathbf{A}_h$ is shared across all frequency modules and all K/V mappings; in the separate setting each module has its own independent projection matrix. The difference in results between using *shared* and *separate* projections is relatively small, though the *separate* version exhibits less stable behavior in the super-resolution setting. More critically, the non-shared design is highly inefficient, expanding the model to 63.6M parameters. Given the similar performance and the large gap in model size, the shared frequency-projection module offers the most practical and efficient configuration.

**Effects of scaled log-spectral loss**  In Table 7, we assess whether auxiliary spectral losses provide benefits. Using perceptual loss alone leads to less stable optimization, whereas adding any spectral term consistently improves performance, confirming the importance of spectral constraints. Among regression-based losses, the $\ell_1$ loss on complex STFT components (magnitude, real, imaginary) outperforms magnitude-only variants by better preserving signal fidelity. Replacing $\ell_1$ with $\ell_2$ slightly degrades performance, likely due to oversmoothing. These results indicate that perceptual loss is essential for high-level quality but must be paired with an appropriate spectral objective.

We next compare log- and scaled log-spectral formulations. A plain log1p loss behaves similarly to $\ell_1$ because typical spectral distances are far below 1, keeping its gradient near 1. The proposed scaled log-spectral loss provides additional gains by adjusting gradient magnitude according to the target spectrum: suitable scale values balance well-aligned and poorly aligned regions, whereas overly small scales collapse gradients and damage performance. Removing perceptual loss noticeably harms both $\ell_1$ and s-log1p, and in this setting $\ell_1$ remains more stable, showing that s-log1p is not effective as a standalone objective. The best overall results arise when $w_{tf}$ is adaptively derived from the target magnitude, demonstrating that the proposed magnitude-adaptive scaling offers the most reliable trade-off between fine spectral detail and global coherence.

## 6. Conclusion

We presented TF-Restormer for speech restoration under the xSFI setting. A query-based asymmetric encoder–decoder analyzes the observed input bandwidth and reconstructs missing high-frequency components via extension queries with band-partitioned cross-attention, and a rate-conditional information flow lets a single model operate across arbitrary input–output rate pairs without redundant resampling.

To stabilize unified training across heterogeneous rates and distortions, we paired a shared multi-scale SFI-STFT discriminator with the s-log1p loss whose transition-point parameter $w_{tf}$ is set by the per-bin source magnitude, exploiting the near-perfect heteroscedasticity between spectral error and source magnitude to equalize the learning signal across frequency bins. Extensive evaluations on denoising, SSR at multiple rate gaps, and GSR confirm balanced gains in both signal fidelity and perceptual quality.

A formal subjective listening study (Appendix N) remains the principal direction for future work; overall, TF-Restormer demonstrates that explicit xSFI modeling, complemented by an adaptive spectral objective and a rate-agnostic adversarial signal, is a practical path to universal restoration.

## Acknowledgements

This work was partly supported by Institute of Information & communications Technology Planning & Evaluation(IITP) grant funded by the Korea government(MSIT)(RS-2022-II220621, Development of artificial intelligence technology that provides dialog-based multi-modal explainability) and National Research Foundation of Korea(NRF) grant funded by the Korea government(MSIT)(RS-2026-25470024, Research on Spatial Audio Reasoning with Large Language Models across Diverse Microphone Arrays).

## Impact Statement

We use only public speech corpora and collect no new personal data. We do not attempt speaker re-identification, and we do not redistribute raw audio. Aware of potential misuse (e.g., covert monitoring), we will apply access controls and intended-use restrictions and require legal compliance for any release.

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

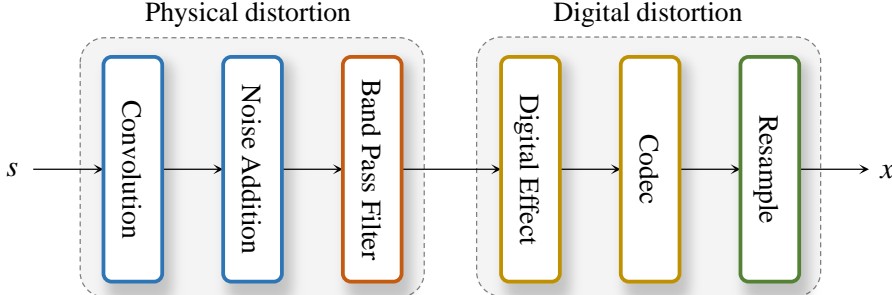

*Figure 5.* **Noisy-distorted speech input simulation pipeline.** The simulation procedure is partitioned to physical distortion and digital distortion.

## A. Details of Training Procedure

### A.1. Simulation of training dataset

**Clean speech source**    The model is trained with VCTK training set. VCTK corpus (Yamagishi et al., 2019) is a multi-speaker English corpus containing 110 speakers with different accents. We split it into a training part VCTK-Train and a testing part VCTK-Test. The version of VCTK we used is 0.92. To follow the data preparation strategy of previous restoration studies Liu et al. (2022b), only the *mic1* microphone data is used for experiments, and *p280* and *p315* are omitted for the technical issues. For the remaining 108 speakers, the last 8 speakers, *p360,p361,p362,p363,p364,p374,p376,s5* are split as test set VCTK-Test for super-resolution subtask. Within the other 100 speakers, *p232* and *p257* are also excluded because they are used in the test set VCTK-SR(noisy-distorted) and VCTK+DEMAND datasets. Therefore, the remaining 98 speakers are used as training data.

**Simulation pipeline**    To simulate input signal for training, we randomly applied the various distortions based on the pipeline as shown in Figure 5. In particular, we sequentially applied physical and digital distortions. The physical distortions include convolution of transfer function mainly caused by reverberation at indoor environment. We used RIR samples from DNS dataset (Reddy et al., 2020). Note that we compensated time-delay effect from the convolution by applying direct component of RIR to the corresponding target speech signal. Then, as a second physical distortion, we added various background and interfering noises using noise samples (Reddy et al., 2020) and simulated colored gaussian noise. Each noise source is independently applied with signal-to-noise (SNR) ratio ranging from 0 to 20 dB. Then, as a final stage of physical distortion, we applied band pass filtering (BPF) to account for the recording condition of microphone such as occlusion, hardware properties, in this study, we mainly considered occlusion effect for the simulation. Also, to remove the phase distortion from the BPF, we applied as zero-phase filtering because the model does not need to consider these effect, only to make the learning process complicated. As a final step for physical simulation, we randomly scaled the level of signals from -35 to -15 dB Full Scale (dBFS). We also scaled the speech sources along with the corresponding input.

Then, three kinds of digital distortions were simulated in sequence. We randomly applied audio clipping, crystalizer, flanger, and crusher as digital effects, each introducing characteristic nonlinear saturation, spectral over-enhancement, comb filtering, or quantization noise (detailed parameter ranges are summarized in Table 8). Afterward, digital codec compression was applied to emulate transmission artifacts, using either MP3 or OGG (Vorbis/Opus) encoding. Finally, the processed signals were randomly downsampled to 8 or 16 kHz to simulate low-bandwidth recording and communication scenarios.

### A.2. Training Details for Unified Model

For pretraining, TF-Restormer was optimized with a batch size of 2 on a single NVIDIA RTX 6000 Ada 48GB GPU using AdamW (Loshchilov & Hutter, 2019). Pretraining was run for 200,000 steps on the VCTK dataset with 3-second utterances. Adversarial training was then applied for an additional 200,000 steps. We used a learning rate of 2.0e-4 with betas (0.9, 0.995), applying a decay of 0.9 every 10,000 steps after 100,000 steps during pretraining, and every 10,000 steps during adversarial training.

Both stages used a 5,000-step linear warm-up for the generator. In adversarial training, the discriminator was updated twice per generator step (without warm-up), using AdamW with betas (0.8, 0.999). Following multi-scale STFT discriminator

*Table 8.* Training-time distortions with probabilities and parameter ranges.

| Augmentation | Prob. | Param. name | Range / Values | Notes |
|---|---|---|---|---|
| RIR convolution | 0.50 | - | - | direct-path delay compensated |
| Sample Noise | 1.00 | SNR (dB) | $[0, 20]$ | from DNS dataset |
| Colored Gaussian Noise | 1.00 | SNR (dB) | $[0, 20]$ | |
| | | exponent $\beta$ | $[0.75, 1.5]$ | |
| Band-limiting (BPF) | 0.5 | $f_1$ (Hz) | $[500, 1500]$ | zero-phase |
| (occlusion FIR) | | $f_2$ (Hz) | $f_1 + [200, 500]$ | transition band upper edge |
| | | cut_gain | $(0.1, 0.3)$ | stopband gain, applied as $g^\beta$ |
| | | $\beta$ | $[0.25, 1.00]$ | (thus effective stopband $\approx [0.22, 0.55]$) |
| | | taps | odd in $[31, 61]$ | `firwin2`, $f_s$=16k ($f_N$=8k) |
| Clipping | 0.5 | level (dB) | $[-15, 0]$ | hard clipping threshold |
| Crystalizer | 0.15 | intensity | $[1, 4]$ | spectral "sharpening" |
| Flanger | 0.05 | depth | $[1, 5]$ | short-delay comb filtering |
| Crusher (bit-depth) | 0.10 | bits | $[1, 9]$ | quantization/aliasing |
| Codec (any) | 0.30 | — | — | one of the following |
| MP3 | | bit rate (kbps) | $[4, 16]$ | variable bit-rate sampled uniformly |
| OGG | | encoder | vorbis, opus | random choice |
| Frequency Masking | 1.00 | $F_{bw}$ (freq. bins) | $[0, 10]$ | |
| | | # masks | $[0, 3]$ | set to $[0, 1]$ in adversarial training |
| Time Masking | 1.00 | $T_{dur}$ (frames) | $[0, 10]$ | |
| | | # masks | $[0, 2]$ | set to $[0, 1]$ in adversarial training |
| Downsample | 1.00 | target $f_s$ | {8k (0.25), 16k (0.75)} | |

designs (Défossez et al., 2023), we employed our multi-scale SFI-STFT discriminator with STFT window sizes of [20, 40, 60, 80, 100] ms to capture spectral details at multiple resolutions.

Across all ablation variants, validation loss plateaued around 60k–70k steps, and no architecture exhibited signs of overfitting. We observed that checkpoint selection within this plateau region led to negligible performance differences, indicating that the chosen training length is sufficient for convergence and provides a fair comparison across variants.

### A.3. Training Details for dedicated model

**VCTK+DEMAND** Since noise reduction does not require generating new speech components, prior work has shown that standard supervised learning is often sufficient. Therefore, in the fine-tuning stage we use small weights in adversarial loss ($\lambda_g = 0.001$ and $\lambda_{fm} = 0.01$ and the human-feedback perceptual loss ($\lambda_{hf} = 10^{-5}$). The model is trained to perform pure denoising following the standard VCTK+DEMAND training partition. The input and output sampling rates are both fixed to 16 kHz, and thus no extension queries are used in this setting.

**VCTK for Super-resolution** For super-resolution, the model is trained under the same protocol as conventional SR systems, using clean low-band inputs as supervision. Consequently, during adversarial fine-tuning the human-feedback loss is again applied with a small weight ($\lambda_{hf} = 10^{-5}$), as the task primarily focuses on recovering missing high-frequency content.

*Table 9.* Distortions for constructing the VCTK-SR(noisy-distorted) evaluation set, with probabilities and parameter ranges.

| Augmentation | Prob. | Param. name | Range / Values | Notes |
|---|---|---|---|---|
| RIR convolution | 0.50 | - | - | direct-path delay compensated |
| Sample Noise | 1.00 | SNR (dB) | $[5, 20]$ | from DNS dataset |
| Colored Gaussian Noise | 1.00 | SNR (dB) | $[5, 20]$ | |
| | | exponent $\beta$ | $[0.75, 1.5]$ | |
| Band-limiting (BPF) | 0.20 | $f_1$ (Hz) | $[2000, 4000]$ | zero-phase |
| (occlusion FIR) | | $f_2$ (Hz) | $f_1 + [200, 500]$ | transition band upper edge |
| | | cut_gain | $(0.1, 0.3)$ | stopband gain, applied as $g^\beta$ |
| | | $\beta$ | $[0.25, 0.75]$ | (thus effective stopband $\approx [0.22, 0.55]$) |
| | | taps | odd in $[31, 61]$ | `firwin2`, $f_s$=16k ( $f_N$=8k ) |
| Clipping | 0.20 | level (dB) | $[-10, 0]$ | hard clipping threshold |
| Crystalizer | 0.10 | intensity | $[1, 2]$ | spectral "sharpening" |
| Flanger | 0.05 | depth | $[1, 3]$ | short-delay comb filtering |
| Crusher (bit-depth) | 0.10 | bits | $[1, 5]$ | quantization/aliasing |
| Codec (any) | 0.25 | — | — | one of the following |
| MP3 | | bit rate (kbps) | $[16, 64]$ | variable bit-rate sampled uniformly |
| OGG | | encoder | vorbis, opus | random choice |
| Frequency Masking | 1.00 | $F_{bw}$ (freq. bins) | $[0, 5]$ | |
| | | # masks | $[0, 1]$ | |
| Time Masking | 1.00 | $T_{dur}$ (frames) | $[0, 5]$ | |
| | | # masks | $[0, 1]$ | |

# B. Simulation of VCTK Noisy Distorted Input

The noisy-distorted input from VCTK testset in Table 3 was generated by corrupting clean VCTK utterances with additive noise from DEMAND (Thiemann et al., 2013) and colored Gaussian noise, RIR samples from RWCP (Nakamura et al., 2000) and AIR (Jeub et al., 2009) for reverberation, and distortions such as clipping and band-limiting. Additional digital effects including audio codecs (MP3, OGG) were applied before resampling to various rates (8-48 kHz). This simulation aligns the training pipeline while maintaining samples partitioning of speech, noise, and RIR sources. As a result, the average SDR of input data (in case of $f_E$ =16kHz) is 2.11dB from 2937 utterances. 998 utterances (about 34%) are below SDR=0 dB and 234 utterances (about 8%) are below SDR=-5 dB.

The details of parameter range are summarized in Table 9.

*Table 10.* Comparison of the model size and RTF. RTF is calculated on NVIDIA RTX 4090. [†]We utilized pretrained models from open implementation code from UNIVERSE++ (Scheibler et al., 2024). [‡]The model size of FINALLY includes WavLM whose model size is 358M.

| Model | Model Size (M) | $f_E \to f_D$ (kHz) | MACs(G) | RTF |
|---|---|---|---|---|
| VoiceFixer | 70.3 | $44.1 \to 44.1$ | 12.9 | 0.010 |
| StoRM | 55.1 | $16 \to 16$ | 156.4 | 0.520 |
| UNIVERSE[†] | 46.4 | $16 \to 16$ | 36.9 | 0.014 |
| UNIVERSE++[†] | 84.2 | $16 \to 16$ | 36.9 | 0.015 |
| FINALLY[‡] | 454.0 | $16 \to 48$ | – | – |
| TF-Locoformer | 14.9 | $16 \to 16$ | 246.9 | 0.025 |
| | | $48 \to 48$ | 731.6 | 0.088 |
| TF-Restormer | 30.1 | $8 \to 16$ | 240.8 | 0.009 |
| | | $8 \to 44.1$ | 308.4 | 0.017 |
| | | $16 \to 16$ | 440.9 | 0.034 |
| | | $16 \to 48$ | 518.7 | 0.053 |
| TF-Restormer-*streaming* | 19.0 | $8 \to 16$ | 114.7 | 0.012 |
| | | $8 \to 44.1$ | 138.1 | 0.018 |
| | | $16 \to 16$ | 214.5 | 0.035 |
| | | $16 \to 48$ | 242.0 | 0.049 |

## C. Details of Model Configuration

For TF-Restormer, $C_E$ and $B_E$ for encoder are set to 128 and 6 while $C_D$ and $B_D$ are set to 64 and 3. The kernel size in ConvFFN and the number of heads in MHSA/MHCA are commonly set to $K = 7$ and $H = 4$, respectively. For frequency projection layer, $F_{\text{proj}}$ is set to 512.

For offline TF-Restormer, each input mixture is normalized by dividing it by its standard deviation and the enhanced output is rescaled by the same factor. For streaming version of TF-Restormer, two mamba blocks are used in the time module with $d_{\text{state}} = 16$, causal Conv1D kernel size 3 with expansion factor 4. For streaming version, we still use the non-causal Conv2D layer for input and output projection for robust restoration, therefore the latency increases by two frames, total latency of 80ms (40 ms window, 20 ms hop). Overall, the model size of TF-Restormer is 30.1M for offline mode and 19.0M for streaming mode, which are smaller sizes compared to the existing models.

## D. Comparison of the model size and RTF

In Table 10, we compare model size and multiply-accumulate operations (MACs) for a 1-second-long input using *ptflops* package[4]. We also measure real-time factor (RTF) measured on 4-second-long samples with an NVIDIA RTX 4090. Conventional models operate at fixed input–output sampling rates, which results in fixed MACs regardless of the task configuration. In contrast, TF-Restormer adapts its computation depending on the input and output rates $f_E$ and $f_D$.

Among baselines, StoRM requires 50 diffusion steps, leading to very high MACs and RTF despite its moderate model size. UNIVERSE and UNIVERSE++ reduce the number of steps (8 by default in the open implementation), which lowers the runtime cost compared to StoRM, but their model sizes remain relatively large and the diffusion process cannot be adapted for streaming, representing a fundamental limitation. TF-Locoformer, built on a dual-path design, involves higher computational complexity but benefits from effective parallelism, so its RTF is not as large as its MACs might suggest; its parameter size is also smaller than most diffusion- or vocoder-based systems.

Our proposed TF-Restormer also follows a dual-path formulation, so the raw MACs are relatively large. Nevertheless, RTF remains low in practice, comparable to or even faster than prior dual-path models. Crucially, TF-Restormer optimizes computation according to the input and output sampling rates: for instance, in the $8 \to 16$ kHz setting, redundant high-frequency processing is skipped, yielding a very low RTF. Also, the streaming variant maintains consistently low RTF while preserving accuracy, demonstrating its suitability for real-time applications.

**Computational saving from observed-band encoding.** The encoder operates on the observed input band ($F_E$) only, while symmetric models resample to the full output band ($F_D$). The relative computational saving in the encoder portion is approximately $(F_D - F_E)/F_D$, yielding 50%, 67%, and 82% reductions for $8 \to 16$, $8 \to 24$, and $8 \to 44.1$ kHz settings, respectively (Table 10).

---

[4]https://github.com/sovrasov/flops-counter.pytorch

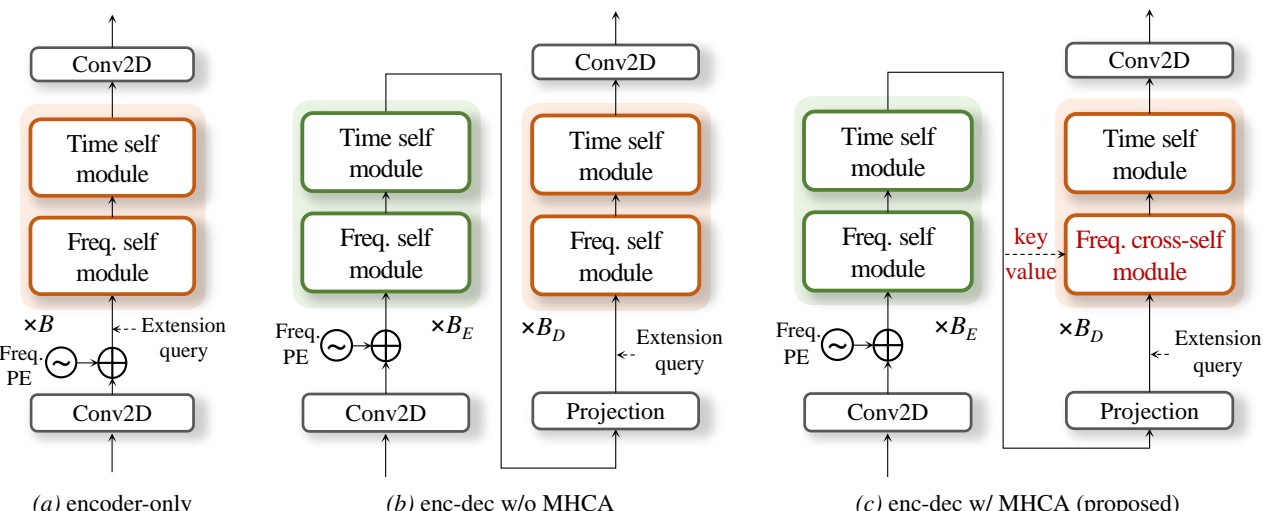

*Figure 6.* **Unit modules in TF-Encoder and TF-Decoder.** The (a) time module is based on MHSA with RoPE while (b) the frequency encoder module is based on MHSA with frequency projection layer. (c) The frequency decoder module utilize MHCA based on key/value from the encoder features

## E. Illustration of encoder-decoder structure ablation

Figure 6 provides detailed comparisons of the three decoder designs considered in our ablation.

**(a) Decoder-only.** This variant directly inserts extension queries into the decoder without an encoder counterpart. The decoder therefore bears the full burden of modeling both the observed input band and the missing high-frequency bands, resulting in heavier computation and weaker inductive bias from the input.

**(b) Encoder-decoder without MHCA.** Here the encoder first analyzes the input bandwidth, and the decoder has the same internal structure as the encoder but receives projected extension queries. Although this design separates analysis and reconstruction, the decoder relies only on self-attention within the extended sequence, and does not explicitly exploit encoder features for reconstruction.

**(c) Encoder-decoder with MHCA (proposed).** In our final design, the decoder additionally uses a frequency cross-self module, where encoder outputs serve as key-value inputs for cross-attention while extension queries act as queries. This enables direct conditioning of high-frequency synthesis on encoder features, while the self-attention within the decoder refines spectral structure among extended bins. As a result, the encoder specializes in processing the observed input, and the lightweight decoder focuses on plausible high-frequency generation guided by encoder information.

These illustrations highlight how the proposed encoder-decoder with MHCA achieves a clear division of labor: the encoder concentrates on input-bandwidth analysis, and the decoder selectively extends spectral content with cross-conditioning, leading to better efficiency and stability compared to the other two designs.

*Table 11.* Ablation study on input/output sampling-rate distributions to evaluate the robustness of extension-query training. Grey rows denote the default distribution used for baseline models.

| $f_E$ (Hz) | | $8 \rightarrow 16$kHz | | | $8 \rightarrow 44.1$kHz | | | $16 \rightarrow 48$kHz | | |
|---|---|---|---|---|---|---|---|---|---|---|
| 8k | 16k | LSD$^\downarrow$ | MCD$^\downarrow$ | NISQA$^\uparrow$ | LSD$^\downarrow$ | MCD$^\downarrow$ | NISQA$^\uparrow$ | LSD$^\downarrow$ | MCD$^\downarrow$ | NISQA$^\uparrow$ |
| Input | | 3.36 | 11.38 | 1.91 | 3.64 | 11.47 | 1.91 | 3.48 | 11.37 | 1.73 |
| 0.01 | 0.99 | 1.48 | 5.83 | 4.01 | 1.46 | 7.12 | 4.41 | **1.15** | **2.82** | 4.55 |
| 0.10 | 0.90 | 1.21 | 2.82 | 4.48 | 1.24 | 3.20 | 4.46 | 1.17 | 2.85 | **4.56** |
| 0.25 | 0.75 | 1.16 | 2.78 | 4.49 | 1.18 | 3.08 | **4.52** | 1.18 | 2.86 | 4.54 |
| 0.50 | 0.50 | **1.14** | **2.74** | **4.50** | 1.17 | **3.05** | 4.52 | 1.20 | 2.89 | 4.53 |

*(a)* ablation of training $f_E$ distribution.

| $f_D$ (Hz) | | | | $8 \rightarrow 16$kHz | | | $8 \rightarrow 24$kHz | | | $8 \rightarrow 44.1$kHz | | | $16 \rightarrow 48$kHz | | |
|---|---|---|---|---|---|---|---|---|---|---|---|---|---|---|---|
| 16k | 24k | 44.1k | 48k | LSD$^\downarrow$ | MCD$^\downarrow$ | NISQA$^\uparrow$ | LSD$^\downarrow$ | MCD$^\downarrow$ | NISQA$^\uparrow$ | LSD$^\downarrow$ | MCD$^\downarrow$ | NISQA$^\uparrow$ | LSD$^\downarrow$ | MCD$^\downarrow$ | NISQA$^\uparrow$ |
| Input | | | | 3.36 | 11.38 | 1.91 | 3.49 | 11.60 | 1.91 | 3.64 | 11.47 | 1.91 | 3.48 | 11.37 | 1.73 |
| 0.49 | 0.49 | 0.01 | 0.01 | **1.15** | **2.73** | **4.51** | 1.17 | 3.05 | **4.52** | 1.36 | 4.01 | 4.49 | 1.51 | 6.83 | 4.01 |
| 0.45 | 0.45 | 0.05 | 0.05 | **1.15** | 2.74 | **4.51** | **1.16** | 3.05 | 4.51 | 1.20 | 3.16 | 4.49 | 1.27 | 3.00 | 4.46 |
| 0.40 | 0.40 | 0.10 | 0.10 | 1.16 | 2.76 | 4.50 | **1.16** | **3.04** | 4.51 | 1.21 | 3.15 | 4.49 | 1.23 | 2.94 | 4.49 |
| 0.25 | 0.25 | 0.25 | 0.25 | 1.16 | 2.78 | 4.49 | 1.18 | 3.08 | **4.52** | **1.18** | **3.08** | **4.52** | **1.18** | **2.86** | **4.54** |

*(b)* ablation of training $f_D$ distribution

## F. Extension Query Under Imbalanced Sampling-Rate Distributions

Table 11 analyzes whether extension-query tokens become undertrained when certain sampling rates appear too infrequently during training. For the input-rate ablation (top subtable), performance remains nearly identical to the balanced baseline as long as 8 kHz inputs constitute at least 10% of the data. The differences across {0.10, 0.25, 0.50} distributions are minimal in all target-rate settings, and even the best scores often occur at moderately imbalanced ratios. Noticeable degradation appears only in the extreme 1% case, where the model sees almost no examples of low-band inputs; this leads to modest but consistent drops, particularly for the largest gap ($8 \rightarrow 44.1$ kHz).

A similar pattern is observed for the output-rate ablation (bottom subtable). When high-frequency target rates (44.1 kHz or 48 kHz) have extremely low probability (1–5%), reconstruction quality decreases for those specific targets, as seen in elevated LSD/MCD values. However, once each output rate is represented with a reasonable frequency (around 10% or more), the performance aligns closely with the uniformly balanced case, and the differences across distributions remain small.

Overall, these results show that extension-query undertraining affects performance only under highly skewed sampling-rate distributions. TF-Restormer remains robust as long as each rate appears with moderate frequency, and balanced or mildly imbalanced settings exhibit negligible differences from the baseline.

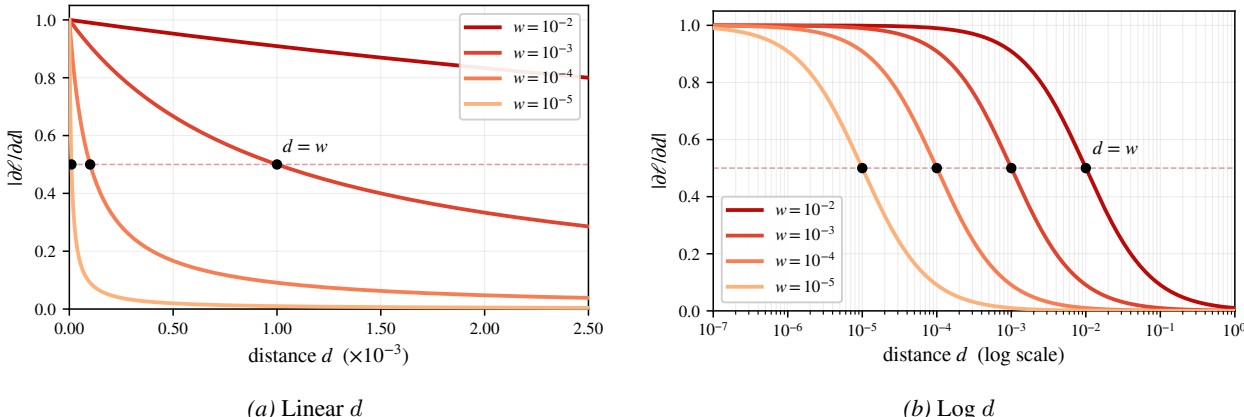

*(a)* Linear $d$                                                    *(b)* Log $d$

*Figure 7.* Gradient magnitude $|\partial\ell/\partial d| = w/(w + |d|)$ of the proposed s-log1p loss for several scale factors $w$. **(a)** Linear distance axis emphasizes the decay rate — smaller $w$ values make the gradient drop faster, increasing sensitivity to fine spectral deviations. **(b)** Logarithmic distance axis reveals scale-invariance: the gradient is a pure function of $d/w$, so all four curves are decade-shifted copies of a single sigmoid centered at $d = w$ (black dots, gradient $= 1/2$); $w$ acts purely as a horizontal stretch in $(\log d)$-space.

## G. Effects of scaled log-spectral loss

Figure 7 illustrates the gradient behavior of the proposed scaled log-spectral loss, defined as $\partial\ell/\partial d = w/(d + w)$ where $d = |y - s|$ denotes the spectral distance and $w$ is a scale factor. Unlike conventional $\ell_1$ or $\ell_2$ criteria, whose gradients are either constant regardless of error magnitude ($\ell_1$) or increase proportionally with larger errors ($\ell_2$), the proposed formulation yields gradients that are strongest near $d \approx 0$ and gradually diminish once $d$ exceeds $w$. This mechanism emphasizes regions where the spectrum is already well-aligned, thereby preserving fine details, while suppressing unstable updates from heavily corrupted regions. The figure shows that when $d = w$, the gradient magnitude stabilizes at $0.5$, providing a natural balance between emphasizing accurate components and de-emphasizing severely mismatched ones. As $w$ decreases, the loss becomes more sensitive to smaller deviations, further suggesting subtle spectral structures that are otherwise neglected in conventional losses.

## H. Loss Family Comparison

Section 4.1 introduces s-log1p as the simplest single-parameter form satisfying the three design criteria (i)–(iii). To make that claim concrete, this appendix situates s-log1p within the broader robust-loss family and examines each member under two parameterizations: the *original* forms used in the robust-estimation literature (Table 12, columns 2–5) and their *peak-normalized* variants (column 6). Table 12 consolidates both, and Figure 8 visualizes the peak-normalized gradient profiles so that the remaining differences reflect shape rather than scale.

**Original forms and the scale-invariance gap**    Columns 2–3 of Table 12 list the standard formulations from the robust-estimation literature. The gradient magnitudes partition into three asymptotic regimes: *unbounded* ($\ell_2$, whose gradient diverges and remains susceptible to outliers), *bounded* ($\ell_1$, Huber, and Charbonnier, whose gradient saturates to a non-zero constant), and *descending* (s-log1p, Cauchy, Geman–McClure, and Welsch, whose gradient decays to zero). Criterion (i) of Section 4.1 selects the descending regime. Within these formulations, criterion (ii) — a $w$-independent unit gradient peak — is satisfied *automatically* only by $\ell_1$, Huber, Charbonnier, and s-log1p. Cauchy, Geman–McClure, and Welsch carry peak heights of $w/2$, $\propto 1/w$, and $\propto w$ respectively, so in their original parameterization the per-bin scaling $w_{tf} = \mathrm{E}_t[S_{m,tf}]$ adopted in Section 4.1 would translate into systematically different gradient magnitudes for low- and high-energy frequency bins. Restoring criterion (ii) for those three losses therefore requires an *explicit* per-family rescaling, which is the contrast underlying the next paragraph.

**Peak-normalized variants**    The scale-invariance gap is repaired by multiplying the gradient of Cauchy, Geman–McClure, and Welsch by a per-family scalar so that $\sup|\partial\ell/\partial d| = 1$ independent of $w$, and integrating back to obtain the loss $\tilde{\ell}(d)$ listed in column 6: $w\log(1 + d^2/w^2)$ for normalized Cauchy, $\frac{8\,d^2}{3\sqrt{3}\,w(d^2+w^2)}$ for normalized Geman–McClure, and

*Table 12.* Robust-loss family compared along design criterion (ii). Columns 2–3 list the original loss $\ell(d)$ and its gradient magnitude $|\partial\ell/\partial d|$. Column 4 reports the supremum of the gradient magnitude (*peak height*), and column 5 marks whether that peak equals 1 *independent of* the scale parameter $w$ — the scale-invariance condition required by criterion (ii) of Section 4.1. Column 6 lists the peak-normalized variant $\tilde{\ell}(d)$ obtained by rescaling the original gradient by a per-family multiplier ($2/w$ for Cauchy, $16/(3\sqrt{3}\,w)$ for Geman–McClure, $\sqrt{2e}/w$ for Welsch) so that $\sup|\partial\tilde{\ell}/\partial d| = 1$ and integrating back; $\ell_1$, Huber, Charbonnier, and s-log1p already have unit peak by construction. $\ell_2$ is unbounded and is retained only as a reference point.

| Loss | Original $\ell(d)$ | $\left\|\dfrac{\partial\ell}{\partial d}\right\|$ | Peak height | Scale-inv? | Peak-normalized $\tilde{\ell}(d)$ |
|---|---|---|---|---|---|
| $\ell_2$ | $\dfrac{1}{2}\,d^2$ | $\|d\|$ | (diverges) | — | — |
| $\ell_1$ | $\|d\|$ | $1$ | $1$ | ✓ | intrinsic |
| Huber | $\begin{cases}\dfrac{d^2}{2w}, & \|d\| \leq w \\ \|d\| - \dfrac{w}{2}, & \|d\| > w\end{cases}$ | $\min\left(\dfrac{\|d\|}{w}, 1\right)$ | $1$ | ✓ | intrinsic |
| Charbonnier | $\sqrt{d^2 + w^2} - w$ | $\dfrac{\|d\|}{\sqrt{d^2 + w^2}}$ | $1$ (asymp.) | × (slope $1/w$) | intrinsic (asymp.) |
| Cauchy | $\dfrac{w^2}{2}\log\left(1 + \dfrac{d^2}{w^2}\right)$ | $\dfrac{w^2\,\|d\|}{w^2 + d^2}$ | $\dfrac{w}{2}$ | × | $w\log\left(1 + \dfrac{d^2}{w^2}\right)$ |
| Geman–McClure | $\dfrac{d^2}{2\,(d^2 + w^2)}$ | $\dfrac{w^2\,\|d\|}{(d^2 + w^2)^2}$ | $\propto \dfrac{1}{w}$ | × | $\dfrac{8\,d^2}{3\sqrt{3}\,w\,(d^2 + w^2)}$ |
| Welsch | $\dfrac{w^2}{2}\left(1 - e^{-d^2/w^2}\right)$ | $\|d\|\,e^{-d^2/w^2}$ | $\propto w$ | × | $\dfrac{w\sqrt{2e}}{2}\left(1 - e^{-d^2/w^2}\right)$ |
| s-log1p (ours) | $w\log\left(1 + \dfrac{\|d\|}{w}\right)$ | $\dfrac{w}{w + \|d\|}$ | $1$ | ✓ | intrinsic |

$\frac{w\sqrt{2e}}{2}(1 - e^{-d^2/w^2})$ for normalized Welsch. The normalized Cauchy form is particularly instructive: it parallels s-log1p as a $w$-scaled logarithm, with only $|d|/w$ replaced by $(d/w)^2$ inside the log. The two forms therefore sit one symbolic step apart at the formula level, yet differ qualitatively in their gradient behavior at the origin — a distinction that becomes the subject of the next paragraph. Figure 8 plots all seven peak-normalized gradients on the scale-invariant axis $|d|/w$ so that the comparison reflects shape alone.

**Remaining structural difference: cusp vs. bilateral saturation** Once peak height is factored out, the descending forms diverge only in their behavior near the origin. The s-log1p gradient retains an $\ell_1$-like *cusp* ($\partial\ell/\partial d \to 1$ as $d \to 0$), so well-aligned regions still receive a strong refinement signal. Normalized Cauchy, Geman–McClure, and Welsch instead exhibit *bilateral saturation*, with the gradient vanishing at both $|d| \to 0$ and $|d| \to \infty$. The latter is not categorically inferior: it concentrates learning capacity on the marginally-mispredicted region near $|d| \sim w$, at the cost of attenuating gradient signal where the prediction is already well-aligned but fine spectral detail can still benefit from continued refinement. Which behavior better suits spectral regression, where fine-grained alignment is itself the target, remains an open empirical question that we leave for future investigation. The narrower claim we make here is structural: among the descending forms in Table 12, s-log1p is the only one that satisfies all three criteria of Section 4.1 in its original parameterization, with $w$ as its single transition-point hyperparameter; the bilateral-saturation alternatives can be aligned with criterion (ii) only by introducing a second per-family rescaling factor. This is the sense in which s-log1p is the simplest single-parameter form within the family.

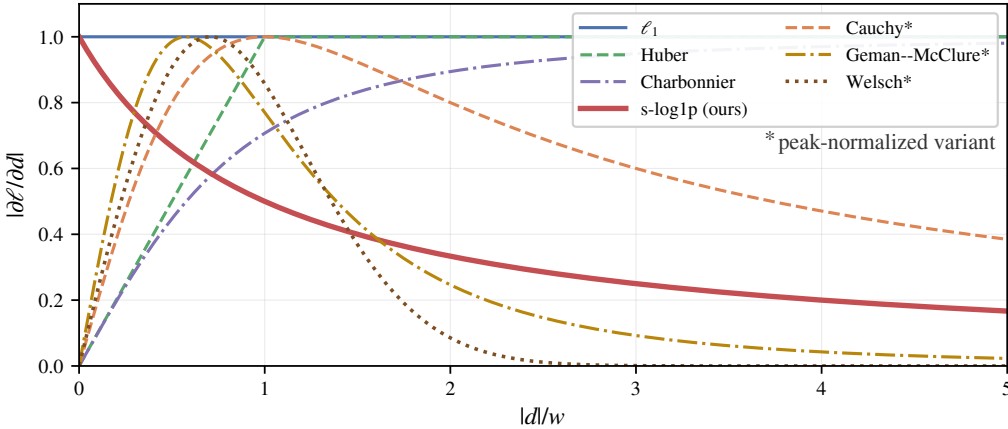

*Figure 8.* Gradient magnitude $|\partial\ell/\partial d|$ as a function of the scale-invariant abscissa $|d|/w$ for the losses in Table 12 ($\ell_2$ is omitted as its gradient diverges). Cauchy, Geman–McClure, and Welsch are shown in their peak-normalized form (column 6 of Table 12); $\ell_1$, Huber, Charbonnier, and s-log1p already have unit peak in their original parameterization. With peak height factored out, the seven curves separate into three structural groups: *bounded* ($\ell_1$, Huber, Charbonnier — gradient saturates to a constant at large $|d|$), *cusp-descending* (s-log1p — gradient $\rightarrow 1$ as $d \rightarrow 0$ and $\rightarrow 0$ as $|d| \rightarrow \infty$), and *bilateral-descending* (Cauchy, Geman–McClure, Welsch — gradient $\rightarrow 0$ at both $|d| \rightarrow 0$ and $|d| \rightarrow \infty$).

## I. Heteroscedasticity of Spectral Errors

We supplement the body-level Figure 4 with per-dataset statistics (Table 13) confirming that the choice of $w_{tf} = \mathrm{E}_t[S_{m,tf}]$ generalizes beyond the single illustrative condition (VCTK-SR(noisy-distorted) 16→48 kHz).

*Table 13.* Per-frequency-bin and per-utterance Spearman correlation between the source magnitude $\mathrm{E}[|S(f)|]$ and the prediction error $\mathrm{E}[|S(f) - \hat{S}(f)|]$, together with coefficient-of-variation (CV) reduction from per-bin magnitude normalization (raw → normalized). The per-bin correlation is near-perfect (0.94–0.99 across all conditions; all $p \approx 0$), and normalizing by source magnitude reduces CV by up to 89%, supporting the magnitude-adaptive $w_{tf}$ choice.

| Dataset | Utt. $\rho$ | Freq. $\rho$ | CV raw → norm. | Reduction |
| --- | --- | --- | --- | --- |
| VCTK+DEMAND | 0.49 | 0.96 | 2.36 → 1.35 | 43% |
| VCTK-SR(noisy-distorted, 8 → 16 k) | 0.50 | 0.94 | 1.62 → 0.42 | 74% |
| VCTK-SR(noisy-distorted, 8 → 44.1 k) | 0.64 | 0.99 | 2.28 → 0.34 | 85% |
| VCTK-SR(noisy-distorted, 8 → 48 k) | 0.58 | 0.98 | 2.65 → 0.28 | 89% |

The near-perfect per-bin correlation across heterogeneous conditions—pure denoising (VCTK+DEMAND) and three super-resolution rate gaps—shows that the magnitude–error proportionality is a structural property of spectral regression rather than a condition-specific artifact. Setting $w_{tf}$ to the per-bin source magnitude therefore equalizes the learning signal across low- and high-energy bins, consistent with variance-stabilizing transformations for heteroscedastic regression.

*Table 14.* Evaluation results on URGENT 2025 (non-blind test set)

| Model(Team) | Model Type | Rank | intrusive signal fidelity | | | | | Semantic fidelity | | | Non-intrusive quality | | |
|---|---|---|---|---|---|---|---|---|---|---|---|---|---|
| | | | PESQ$\uparrow$ | ESTOI$\uparrow$ | SDR$\uparrow$ | MCD$\downarrow$ | LSD$\downarrow$ | sBERT$\uparrow$ | SpkSim$\uparrow$ | CAcc(%)$\uparrow$ | UTMOS$\uparrow$ | NISQA$\uparrow$ | DNSMOS$\uparrow$ |
| Input | - | - | 1.37 | 0.61 | 2.53 | 7.92 | 5.51 | 0.75 | 0.63 | 81.29 | 1.56 | 1.69 | 1.84 |
| baseline | D | 9 | 2.43 | 0.80 | 11.29 | 3.32 | 2.84 | 0.86 | 0.80 | 84.96 | 2.11 | 2.89 | 2.94 |
| Bobbsun | D | 1 | **2.95** | **0.86** | **14.33** | 3.01 | 2.83 | **0.91** | **0.85** | **88.92** | 2.09 | 3.22 | 2.88 |
| USEM(rc) | D | 2 | 2.79 | 0.85 | 13.11 | **2.93** | **2.94** | 0.90 | 0.84 | 88.05 | 2.30 | 3.21 | 3.01 |
| USEM-Flow(rc) | G | - | 1.54 | 0.59 | 4.49 | 6.10 | 3.91 | 0.76 | 0.66 | 69.07 | 1.79 | 2.82 | 2.50 |
| TS-URGENet(Xiaobin) | D+G | 3 | 2.74 | 0.84 | 13.06 | 3.30 | 3.08 | 0.89 | 0.84 | 87.94 | 2.16 | 3.24 | 2.92 |
| alindborg | D+G | 10 | 1.99 | 0.76 | 7.49 | 4.51 | 3.73 | 0.84 | 0.77 | 81.70 | 2.49 | 3.96 | **3.28** |
| wataru9871 | G | 13 | 1.36 | 0.56 | -13.88 | 11.25 | 7.98 | 0.82 | 0.51 | 79.70 | 2.53 | 3.74 | 3.10 |
| TF-Restormer (*original*) | D+G | - | 1.71 | 0.71 | 4.19 | 4.84 | 4.32 | 0.84 | 0.73 | 80.21 | **3.57** | **4.51** | 3.25 |
| TF-Restormer (*dedicated*) | D+G | - | 2.60 | 0.83 | 11.78 | 3.18 | 2.91 | 0.88 | 0.82 | 84.28 | 2.47 | 3.43 | 3.06 |
| TF-Restormer (*dedicated w/ original loss*) | D+G | - | 1.82 | 0.72 | 5.24 | 4.80 | 4.45 | 0.83 | 0.75 | 81.53 | 3.08 | 3.96 | 3.24 |

## J. Evaluation of Dedicated Model on the URGENT Challenge

In the main paper, we reported non-intrusive MOS results on the URGENT blind test set using the unified TF-Restormer model. Since the blind set does not provide clean references, only MOS-based evaluation is possible. For completeness, we additionally train a dedicated URGENT model and report objective fidelity metrics on the non-blind validation set in the challenge (Table 14).

For the dedicated URGENT configuration, we follow the official training recipe. Because the benchmark resamples all inputs to the target sampling rate regardless of their original rate, we remove high-frequency bins with negligible power prior to processing, which reduces redundant computation under this matched-rate setting. Apart from this preprocessing, the model is trained under the same conditions required by the challenge.

It is important to note that the URGENT benchmark emphasizes deterministic, fidelity-oriented enhancement (Sun et al., 2025; Chao et al., 2025; Rong et al., 2025). Intrusive metrics are heavily weighted, and top-ranked systems typically rely on large, deterministic architectures optimized exclusively for matched-rate denoising. In contrast, models based on a generative approach to improve perceptual quality generally achieve lower intrusive scores despite producing more natural listening quality.

Under such conditions, a dedicated TF-Restormer variant trained with the URGENT recipe shows moderate improvement in signal fidelity compared to the unified model; however, its performance remains below that of highly specialized deterministic and multi-stage systems, and its perceptual quality drops significantly. This outcome is expected: the strengths of TF-Restormer—arbitrary input–output sampling-rate handling, frequency-extension mechanisms, perceptually oriented losses, and adversarial training—do not align with the evaluation objectives of URGENT. Consequently, while the unified model yields strong non-intrusive MOS on the blind test set, the dedicated URGENT-trained version does not fully reflect the core advantages of our architecture.

To verify that the perceptual quality drop in the dedicated URGENT-trained variant is primarily due to loss rebalancing rather than the architecture, we additionally trained a dedicated variant with the original (unified) loss configuration. This "dedicated w/ original loss" variant recovers perceptual metrics (UTMOS 3.08, NISQA 3.96) close to the unified model while modestly improving fidelity over the unified baseline (PESQ 1.82 vs. the unified model's 1.71), supporting the interpretation that the URGENT loss rebalancing—not the architecture—drives the perceptual–fidelity trade-off.

*Table 15.* Comparison of TF-Restorer-*streaming* with existing real-time denoising models on the VCTK+DEMAND test set. We report standard enhancement metrics (PESQ, STOI, CSIG, CBAK, COVL) along with model size and MACs.

| Model | Size (M) | MACs (G/s) | PESQ | CSIG | CBAK | COVL | STOI |
|---|---|---|---|---|---|---|---|
| Noisy | - | - | 1.97 | 3.34 | 2.44 | 2.63 | 0.921 |
| NSNet2 (Braun et al., 2021) | 6.2 | 0.43 | 2.47 | 3.23 | 2.99 | 2.90 | 0.903 |
| DCCRN (Hu et al., 2020) | 3.7 | 14.36 | 2.54 | 3.74 | 3.13 | 2.75 | 0.938 |
| FullSubNet+ (Chen et al., 2022a) | 8.7 | 30.06 | 2.88 | 3.86 | 3.42 | 3.57 | 0.940 |
| FRCRN (Zhao et al., 2021) | 10.3 | 12.3 | **3.21** | 4.23 | **3.64** | **3.73** | - |
| DeepFilterNet (Schroter et al., 2022) | 1.8 | 0.11 | 2.81 | 4.14 | 3.31 | 3.46 | **0.942** |
| DeepFilterNet2 (Schröter et al., 2022) | 2.3 | 0.356 | 3.08 | 4.30 | 3.40 | 3.70 | 0.941 |
| TF-Restormer-*streaming* | 19.0 | 214.5 | 2.89 | **4.37** | 3.41 | 3.68 | 0.937 |

## K. Comparison with Conventional Streaming Enhancement Models

We evaluate TF-Restormer-*streaming* against representative real-time denoising models on the VCTK+DEMAND test set (Table 15), using PESQ, STOI, and the composite metrics CSIG, CBAK, and COVL commonly adopted in prior enhancement works. Unlike conventional streaming models, which are trained specifically for denoising under matched conditions, TF-Restormer-*streaming* inherits the full unified restoration and bandwidth-extension objective and is trained to handle reverberation, distortion, and bandwidth mismatch simultaneously. As a consequence, its model size and computational cost are substantially larger than lightweight denoisers such as NSNet2, DCCRN, or DeepFilterNet.

In terms of signal fidelity, specialized denoising models remain strong, with FRCRN and DeepFilterNet2 achieving the highest PESQ, CBAK, or STOI scores. Nevertheless, TF-Restormer-*streaming* attains competitive perceptual quality, achieving the highest CSIG score among all models and CBAK/COVL values close to the best discriminative systems. This is notable given that TF-Restormer-*streaming* is not optimized for denoising alone, but operates as a general-purpose restoration model that simultaneously handles reverberant, noisy, and bandwidth-limited inputs.

Overall, these results show that TF-Restormer-*streaming* is, to our knowledge, the first unified restoration and super-resolution model capable of streaming operation, while still providing signal fidelity comparable to denoising-oriented baselines. This demonstrates the feasibility of extending multi-rate, multi-distortion restoration models to real-time settings without sacrificing robustness.

*Table 16.* Comparison of TF-Restormer with previous models on 2020 DNS testsets in terms of signal fidelity. "No Reverb" subset is only compared as the proposed TF-Restormer is trained to remove reverberation.

| System | PESQ | STOI(%) | SI-SDR(dB) |
|---|---|---|---|
| Noisy | 1.58 | 91.5 | 9.1 |
| FullSubNet (Hao et al., 2021) | 2.78 | 96.1 | 17.3 |
| CTSNet (Li et al., 2021) | 2.94 | 96.2 | 16.7 |
| TaylorSENet (Li et al., 2022) | 3.22 | 97.4 | 19.2 |
| FRCRN (Zhao et al., 2021) | 3.23 | 97.7 | 19.8 |
| MFNet (Liu et al., 2023) | 3.43 | 97.9 | 20.3 |
| USES (Zhang et al., 2023) | 3.46 | 98.1 | 21.2 |
| TF-Locoformer (Saijo et al., 2024) | 3.72 | 98.8 | **23.3** |
| TF-Restormer | 2.83 | 96.4 | 16.1 |

*Table 17.* Comparison of TF-Restormer with previous models on 2020 DNS testsets in terms of perceptual quality (DNS scores). "With Reverb" subset contains reverberation while "No Reverb" subset only involves noise. "D" and "G" denote discriminative and generative methods, respectively.

| Model | Type | With Reverb | | | No Reverb | | |
|---|---|---|---|---|---|---|---|
| | | SIG | BAK | OVRL | SIG | BAK | OVRL |
| Noisy | - | 1.76 | 1.50 | 1.39 | 3.39 | 2.62 | 2.48 |
| Conv-TasNet (Luo & Mesgarani, 2019) | D | 2.42 | 2.71 | 2.01 | 3.09 | 3.34 | 3.00 |
| FRCRN (Zhao et al., 2021) | D | 2.93 | 2.92 | 2.28 | 3.58 | 4.13 | 3.34 |
| SELM (Wang et al., 2024) | G | 3.16 | 3.58 | 2.70 | 3.51 | 4.10 | 3.26 |
| MaskSR (Li et al., 2024) | G | 3.53 | 4.07 | 3.25 | 3.59 | 4.12 | 3.34 |
| AnyEnhance (Zhang et al., 2025) | G | 3.50 | 4.04 | 3.20 | 3.64 | **4.18** | 3.42 |
| GenSE (Yao et al., 2025) | G | 3.49 | 3.73 | 3.19 | 3.65 | **4.18** | **3.43** |
| LLaSE-G1 (Kang et al., 2025) | G | 3.59 | 4.10 | 3.33 | **3.66** | 4.17 | 3.42 |
| UniSE(Yan et al., 2025) | G | **3.67** | 4.10 | **3.40** | **3.67** | 4.14 | **3.43** |
| TF-Restormer | D+G | 3.60 | **4.12** | 3.35 | 3.65 | **4.18** | **3.43** |

# L. Evaluation on DNS Challenge Dataset

We further evaluate TF-Restormer on the 2020 DNS (Reddy et al., 2020) test sets to examine both signal fidelity and perceptual quality. Table 16 compares objective fidelity metrics against models specifically optimized for denoising with DNS training dataset. Since our unified model is trained to remove reverberation as well as noise, we report fidelity scores only on the "No Reverb" subset, where the clean references are aligned with our training objective. Under this setting, the unified TF-Restormer shows lower fidelity than DNS-targeted systems such as MFNet, USES, and TF-Locoformer. This gap is expected, as the unified model (i) is trained solely on VCTK, (ii) actively removes reverberation and other distortions, and (iii) incorporates perceptual objectives that may deviate from strict waveform fidelity. When trained in a DNS-specific manner without adversarial objectives, however, the dedicated TF-Restormer variant matches or surpasses prior systems, achieving competitive PESQ, STOI, and SI-SDR scores.

Table 17 presents perceptual DNSMOS scores on both "With Reverb" and "No Reverb" subsets. Here, discriminative models tend to preserve the input structure and thus achieve relatively conservative perceptual gains, as seen with Conv-TasNet and FRCRN. Generative approaches such as SELM, GenSE, MaskSR, and UniSE, which prioritize perceptual naturalness, obtain noticeably higher OVRL scores. TF-Restormer shows perceptual quality on par with these generative systems across all subsets, despite not being trained exclusively for perceptual enhancement. In both reverberant and non-reverberant conditions, it achieves strong SIG and BAK scores and matches the best OVRL scores among recent models, demonstrating that the proposed architecture can deliver high perceptual quality while maintaining reasonable signal fidelity.

*Table 18.* REVERB-Sim — comparison with prior dereverberation methods. TF-Restormer is evaluated as OOD restoration (no REVERB-specific training). TF-Restormer is the only row evaluated under our unified setup.

| Model | CD↓ | LLR↓ | SNR$_{fw}$ ↑ | PESQ↑ | SRMR↑ |
|---|---|---|---|---|---|
| Input | 3.975 | 0.574 | 3.617 | 1.503 | 3.687 |
| WPE (Yoshioka & Nakatani, 2012) | 3.748 | 0.514 | 4.864 | 1.722 | 4.220 |
| WRN (Llombart et al., 2019) | 3.590 | 0.470 | 4.800 | — | 3.590 |
| GCRN (Hazrati et al., 2013) | 2.534 | 0.325 | 10.753 | 1.934 | 4.861 |
| TCN+SA (Wang et al., 2017) | 2.200 | 0.240 | 13.060 | 2.580 | 5.170 |
| D-MFMVDR (Zhang et al., 2021) | 2.639 | 0.316 | 9.649 | 2.167 | 4.892 |
| DCN (Kothapally & Hansen, 2024) | **2.001** | **0.225** | **13.326** | **2.935** | 5.269 |
| TF-Restormer | 2.880 | 0.489 | 8.139 | 2.825 | **8.004** |

*Table 19.* REVERB-Sim — discriminative baselines (in-domain, trained on REVERB data) and OOD restoration baselines (VoiceFixer, UNIVERSE++). All results are averaged over far / near conditions. discriminative baselines were trained under matched condition. TF-Restormer is evaluated as the same single unified model.

| Model | Fidelity | | | | | Perceptual | | | | |
|---|---|---|---|---|---|---|---|---|---|---|
| | CD↓ | LLR↓ | SNR$_{fw}$ ↑ | PESQ↑ | SRMR↑ | UTMOS | DNSMOS | NISQA | SIG | BAK |
| Input | 3.975 | 0.574 | 3.617 | 1.503 | 3.687 | 1.455 | 1.365 | 1.255 | 1.750 | 1.390 |
| TF-GridNet | **2.110** | **0.216** | **14.458** | 2.825 | 5.063 | 3.325 | 3.135 | 3.310 | 3.435 | 3.945 |
| MP-SENet | 2.692 | 0.278 | 13.622 | 2.811 | 4.960 | 3.225 | 3.190 | 3.500 | 3.450 | 4.035 |
| TF-Locoformer | 2.323 | 0.235 | 14.102 | 2.642 | 4.972 | 3.202 | 3.063 | 3.056 | 3.366 | 3.920 |
| VoiceFixer | 3.708 | 0.946 | 7.992 | 1.278 | 4.938 | 2.858 | 3.088 | 4.072 | 3.388 | 3.927 |
| UNIVERSE++ | 5.213 | 0.901 | 5.731 | 1.220 | 5.124 | 2.370 | 2.835 | 4.145 | 3.150 | 3.835 |
| TF-Restormer | 2.880 | 0.489 | 8.139 | **2.825** | **8.004** | **4.130** | **3.335** | **4.860** | **3.585** | **4.115** |

*Table 20.* REVERB-DNS — constructed by mixing DNS noise at −10 to 0 dB SNR into REVERB-Sim data. TF-Restormer uses the same single model (no re-training); discriminative baselines were re-trained with DNS noise added.

| Model | Fidelity | | | | | Perceptual | | | | |
|---|---|---|---|---|---|---|---|---|---|---|
| | CD↓ | LLR↓ | SNR$_{fw}$ ↑ | PESQ↑ | SRMR↑ | UTMOS | DNSMOS | NISQA | SIG | BAK |
| Input | 5.694 | 0.962 | -0.271 | 1.111 | 2.320 | 1.398 | 1.151 | 1.250 | 1.318 | 1.206 |
| TF-GridNet | **3.654** | **0.604** | **10.101** | 1.867 | 5.696 | 2.587 | 2.961 | 2.905 | 3.285 | 3.824 |
| MP-SENet | 4.347 | 0.780 | 7.807 | 1.493 | 5.217 | 2.072 | 2.641 | 2.607 | 3.082 | 3.416 |
| TF-Locoformer | 4.118 | 0.630 | 8.262 | 1.581 | 5.388 | 2.159 | 2.517 | 2.537 | 2.921 | 3.402 |
| VoiceFixer | 4.370 | 1.070 | 6.150 | 1.350 | 4.989 | 2.227 | 2.852 | 3.534 | 3.177 | 3.660 |
| UNIVERSE++ | 6.123 | 1.115 | 3.418 | 1.190 | 4.473 | 1.909 | 2.322 | 3.245 | 2.697 | 3.184 |
| TF-Restormer | 3.582 | 0.707 | 7.130 | **2.113** | **8.586** | **4.014** | **3.272** | **4.698** | **3.533** | **4.061** |

## M. Evaluation on REVERB Dataset (SimData)

For controlled evaluation under real-world reverberation, we report results on the REVERB-Sim simulated subset (paired references available) and a combined REVERB-DNS condition with added background noise. TF-Restormer is evaluated as an out-of-distribution restoration model (no REVERB-specific re-training); the real-recording subset is reported in Table 4d of the main paper.

Table 18 compares TF-Restormer with prior dereverberation methods (WPE, WRN, GCRN, TCN+SA, D-MFMVDR, DCN) on REVERB-Sim, with baseline scores taken from their respective publications. TF-Restormer trails the best fidelity baseline (DCN) on CD, LLR, and SNR$_{fw}$, but substantially leads on SRMR (8.00 vs. 5.27 for DCN), indicating stronger perceptual dereverberation as a single OOD restoration model.

Table 19 extends the comparison to discriminative baselines trained in-domain on REVERB (TF-GridNet, MP-SENet, TF-Locoformer) and to OOD restoration baselines reproduced under matched conditions (VoiceFixer, UNIVERSE++). In-domain discriminative models retain the fidelity advantage, while TF-Restormer matches the strongest discriminative model on PESQ and uniformly leads on SRMR and all non-intrusive perceptual predictors (UTMOS, DNSMOS, NISQA, SIG, BAK) as a single unified model.

Table 20 reports the harder REVERB-DNS condition, constructed by mixing DNS noise at −10 to 0 dB SNR into REVERB-Sim data; the discriminative baselines were re-trained with the added noise condition while TF-Restormer is reused unchanged. The unified model attains the best PESQ and leads on SRMR as well as on every non-intrusive perceptual predictor, confirming that it degrades gracefully when noise compounds reverberation.

# N. Limitations and Future Works

Although TF-Restormer demonstrates balanced improvements in both signal fidelity and perceptual quality, several limitations remain.

**Hallucination under extreme degradation** When the input is severely degraded, observed-spectrum uncertainty can introduce content or speaker artefacts — a fundamental trade-off between adversarial training's perceptual naturalness (via plausible high-frequency hallucination) and purely supervised objectives' fidelity (often at the cost of oversmoothing).

**Rate-distribution sensitivity** Severely imbalanced exposure to certain sampling rates ($<1\%$) can leave extension-query regions undertrained at the highest frequencies (Appendix F); moderate coverage ($\approx 10\%$) suffices in practice, but this remains a limitation of the arbitrary-rate formulation.

**Language-specific bias** Training on a single-language clean corpus, combined with the perceptual loss's reliance on a pretrained speech model (WavLM), may introduce language-specific bias affecting cross-lingual generalization to underrepresented phonetic patterns.

**Multi-speaker conditions** The current formulation assumes a single target speaker and does not handle multi-speaker conditions such as separation or extraction; given that the TF dual-path architecture was originally proposed for separation, multi-speaker restoration is a natural extension.

**Subjective listening evaluation** A formal subjective listening study (e.g., MUSHRA or AB-preference protocol with $N = 15$–$30$ listeners and $\sim 30$ stimuli per condition) would further strengthen the perceptual claims made via non-intrusive MOS predictors (UTMOS, DNSMOS, NISQA), and is left as future work. As an informal surrogate, we publicly release the full implementation with pretrained weights at `https://github.com/dmlguq456/TF_Restormer` and provide a demo page with audio examples covering all evaluation scenarios at `https://tf-restormer.github.io/demo`, enabling readers to reproduce reported results and perform their own subjective inspection on representative samples.

**Future Works** Future work includes reducing hallucination artifacts under extreme degradations, improving robustness under highly imbalanced sampling-rate distributions, extending the training pipeline to multilingual and more diverse corpora, and integrating the framework with multi-speaker modeling.

