# OpenReview forum: "Query-Based Asymmetric Modeling with Decoupled Input–Output Rates for Speech Restoration"
_ICML.cc/2026/Conference — ICML 2026 regular_

### Official Review · Reviewer_PB8a · 2026-02-19

**Soundness:** 3
**Presentation:** 3
**Significance:** 3
**Originality:** 2
**Overall Recommendation:** 3
**Confidence:** 3

**Summary:**

This paper addresses the challenges of compound distortions and sampling rate mismatch between input and desired output in speech restoration under real-world conditions, proposing **TF-Restormer**, a query-based asymmetric modeling framework. Most existing systems assume fixed and shared input-output sampling rates and rely on external resampling, which limits model generality. By formulating the speech restoration problem under decoupled input-output rates, the authors design an asymmetric encoder-decoder architecture: the encoder focuses on analyzing the observed input bandwidth via a time-frequency dual-path structure, while a lightweight decoder reconstructs missing spectral content through frequency-extension queries. This design enables a single model to operate consistently across arbitrary input-output rate pairs without redundant resampling. Experiments under diverse sampling rates, degradation conditions, and operational modes (including real-time streaming scenarios) demonstrate that TF-Restormer maintains stable restoration behavior and balanced perceptual quality.

**Compliance With Llm Reviewing Policy:**

Affirmed.

**Key Questions For Authors:**

1.  **Mechanism of Frequency Query Slicing:** The paper states that extension queries are sliced from a unified master vector $\tilde{Q}_{ext}$ defined over $[F_{min}, F_{max}]$. Could the authors elaborate on the specific indexing mechanism used when mapping this master vector to specific sampling rates (e.g., 22.05 kHz vs 24 kHz)? Since the number of frequency bins varies non-linearly with sampling rate given a fixed frame duration, does the slicing rely on nearest-neighbor indexing, linear interpolation, or a learned alignment matrix?
2.  **Theoretical Computational Breakdown:** Table 9 demonstrates that MACs vary with input-output rates, suggesting efficiency gains over the "resampling + fixed-rate model" cascade. Rather than additional system-level benchmarks, could the authors provide a theoretical breakdown or formula estimating the computational savings? Specifically, what proportion of the total MACs is attributed to the avoided resampling operations versus the asymmetric decoder design?

3.  **Architectural Barriers for Multi-Speaker Extension:** The limitations section acknowledges the single-speaker assumption. Given that the TF dual-path backbone is historically effective for separation, could the authors technically analyze which specific components prevent multi-speaker extension? Is the limitation primarily due to the single-vector nature of the frequency extension queries, or does the asymmetric encoder-decoder structure lack the necessary capacity to model overlapping spectral masks?

4.  **Clarification of Module Originality:** **Overall, a notable area considered by the study** is the use of frequency extension queries within an asymmetric encoder-decoder. However, given the similarities to existing works like TF-GridNet (dual-path), MAE (masking/queries), and SFI models (Paulus & Torcoli, 2022), I might have overlooked the specific boundaries of novelty. Could the authors provide a concise comparison (either in text or a table) explicitly distinguishing their frequency cross-self module and projection layers from standard attention mechanisms in prior TF models? If this clarification demonstrates a distinct theoretical contribution beyond component integration, I would be willing to reconsider the novelty score positively.

**Limitations:**

yes

**Strengths And Weaknesses:**

### **Strengths:**
1.  **Comprehensive Experimental Evaluation:** The validation spans diverse datasets (UNIVERSE, VCTK, VoxCeleb, URGENT 2025) and tasks (denoising, super-resolution, general restoration). The inclusion of non-intrusive MOS evaluations on the URGENT 2025 blind test set provides credible evidence of robustness in real-world scenarios, surpassing many fixed-rate baselines in perceptual quality.


### **Weaknesses:**

1.  **Limited Component-Level Novelty:** A significant portion of the architecture relies on established modules rather than new theoretical inventions. The TF dual-path backbone resembles TF-GridNet/TF-Locoformer, the streaming mechanism utilizes standard Mamba blocks, and the SFI input formulation follows Paulus & Torcoli (2022). While the *combination* is effective, the lack of novel mathematical formulations in individual layers may limit the theoretical contribution depth and raises questions about performance ceilings compared to purpose-built novel operators.

2.  **Computational Complexity vs. Efficiency:** Although the Real-Time Factor (RTF) is low, the absolute MACs remain high compared to specialized lightweight denoisers (e.g., DeepFilterNet, NSNet2). For extremely resource-constrained edge devices, the dual-path asymmetric structure—even with a lightweight decoder—may still pose a memory and power bottleneck, especially when compared to single-path fixed-rate models.

---

> ### Author Rebuttal · Authors · 2026-03-27
>
> We thank the reviewer for the constructive review, and especially for the willingness to reconsider the novelty assessment. We address each point below.
>
> **W1/Q4: Component-level novelty and module originality comparison.** We thank the reviewer for the opportunity to clarify.
>
> We agree that many building blocks are established. Rather, the contribution lies in a new **conditional information-flow structure** for decoupled input-output rates—*band-partitioned cross-attention for rate-decoupled analysis-synthesis*: (1) observed-band-only encoding, (2) unobserved-band synthesis through extension queries, and (3) band-partitioned cross-attention from missing bands to observed bands.
>
> Prior TF dual-path models with SFI-STFT still apply symmetric frequency processing over all bins under matched rates. In contrast, our main structural differences are:
> (1) the decoder's cross-attention restricts queries to unobserved bins and K/V to observed encoder features, yielding a band-partitioned information flow;
> (2) when f_E=f_D, the MHCA path is bypassed entirely, making the operator conditional on the rate pair rather than always active; and
> (3) the shared K/V projection on the frequency axis imposes a rate-consistent structural prior across physical-frequency bins (validated in Table 6(b)).
>
> While MAE also adopts an asymmetric encoder-decoder at a high level, the analogy is limited in an important respect. MAE reconstructs randomly masked tokens using self-attention, whereas our decoder performs physically structured completion: queries are defined by the missing frequency bands, and they attend only to observed-band encoder features rather than to all tokens uniformly.
>
> | Aspect | TF-models (single-stage) | TF-Restormer Enc | TF-Restormer Dec |
> |---|---|---|---|
> | T-module | MHSA (or LSTM) | MHSA | MHSA |
> | F-module | MHSA (or LSTM) | MHSA + **K/V proj**. | MHCA + MHSA + **K/V proj**. |
> | Freq. range | Resampled full F | **Observed F_E** | **Extended F_D** |
> | F-MHCA query | - | - | **Unobserved only (Q_ext)** |
> | F-MHCA K/V | - | - | Observed only from encoder |
>
> This distinction is also reflected empirically. Table 5 shows the encoder-decoder with MHCA consistently outperforms both encoder-only and w/o-MHCA variants, and the gain scales with the rate gap: LSD improves from 0.89 vs 2.12 at 8→16kHz to 1.01 vs 3.25 at 8→44.1kHz. This scaling behavior suggests the structured cross-attention is increasingly important at larger rate gaps, consistent with a functionally load-bearing design. Furthermore, the size-matched variant (small, 10.9M) outperforms encoder-only (11.6M) with ~41% fewer MACs (89G vs 151G), confirming the improvement comes from architectural design rather than capacity. We believe these distinctions—band-partitioned cross-attention, rate-conditional bypass, and the scaling evidence—together constitute a contribution beyond component integration.
>
> **W2/Q2: Computational analysis.** We agree TF-Restormer is not targeted at ultra-lightweight fixed-rate denoisers. Our target is flexible arbitrary-rate restoration, where conventional pipelines resample to a shared full band and incur redundant computation.
>
> For example, resampling 8k to 44.1k forces processing F=883 bins although only F=161 contain observed signal. More generally, redundancy = (F_D−F_E)/F_D: ~50% (8→16k), ~67% (16→48k), ~82% (8→44.1k). TF-Restormer processes only the observed band in the encoder and handles extension with a lightweight decoder. Table 5 shows: enc-only (415G) vs enc-dec w/MHCA(small) (157G), despite similar model sizes.
>
> **Q1: Frequency query slicing mechanism.** The slicing uses exact integer indexing, with no interpolation. With a fixed 40 ms SFI-STFT window, each frequency bin corresponds to exactly Δ_f = 1/0.04 = 25 Hz regardless of sampling rate. The master vector Q̃_ext is defined over [F_min, F_max] = [161, 961], covering approximately 4–24kHz, where each q_f ∈ R^{C_D} corresponds to the physical band at 25f Hz.
>
> For example, for f_E = 22.05kHz (F_E = 442) and f_D = 24kHz (F_D = 481), the extension queries are Q_ext = [q_443, ..., q_481]. Since 40 ms yields an integer number of samples at all standard rates ({8, 16, 22.05, 24, 32, 44.1, 48} kHz), the bin count is always exactly N_fft/2+1, with no fractional indexing.
>
> **Q3: Multi-speaker extension.** This is not a fundamental impossibility, but it is not directly handled by the current single-output formulation. The present model assumes a single restored target stream. A multi-speaker extension would require either multiple decoder branches or a speaker-conditioned latent decomposition before bandwidth extension. We will clarify that the current limitation is primarily one of formulation scope rather than an architectural barrier.

---

> > ### Author Rebuttal · Reviewer_PB8a · 2026-04-01
> >
> > I want to thank the authors for their comprehensive and thoughtful rebuttal. The additional clarifications, particularly the explicit comparison table and the mathematical breakdown of efficiency, have effectively addressed my primary concerns.
> > Here is a breakdown of my assessment based on the rebuttal:
> > 1. Component-Level Novelty & Module Originality (W1/Q4)
> > I appreciate the authors providing an explicit comparison table between standard TF models and TF-Restormer. The clarification of the conditional information-flow structure—specifically, the band-partitioned cross-attention and the rate-conditional bypass—highlights the architectural novelty beyond mere component integration. The empirical scaling evidence from the ablation study, showing that the structured cross-attention becomes increasingly important at larger rate gaps, further strengthens this argument.
> > 2. Theoretical Computational Breakdown (W2/Q2)
> > The explanation regarding the computational redundancy avoided by not processing unobserved bins, defined by $(F_D - F_E) / F_D$, provides the theoretical justification I was looking for. This clearly demonstrates the efficiency gains of the asymmetric design over standard resampling cascades (saving roughly 50% to 82%, depending on the rate gap), even if the absolute MACs remain higher than those of specialized lightweight denoisers.
> > 3. Mechanism of Frequency Query Slicing (Q1)
> > The explanation of the exact integer indexing using the 40 ms window, which yields a clean $\Delta_f = 25$ Hz per bin, perfectly resolves my concern about non-linear scaling and the potential need for interpolation. It is an elegant and effective design choice that ensures perfect alignment across the standard sampling rates.
> > 4. Multi-Speaker Extension (Q3)
> > I appreciate the candid acknowledgment that the single-speaker constraint is a limitation of the current formulation's scope rather than a strict architectural barrier inherent to the asymmetric design.
> > Conclusion:
> > The authors have directly and satisfactorily answered all of my questions. The theoretical boundaries of the contribution and the mechanics of the query slicing are now much clearer to me. Consequently, my concerns are fully resolved (Option A). I will be adjusting my score to reflect the overall strength of the submission and this robust rebuttal. I highly recommend that the authors integrate these clarifications—especially the architectural comparison table and the explicit explanation of the 25 Hz integer indexing—into the final camera-ready manuscript.

---

> > > ### Author Response · Authors · 2026-04-01
> > >
> > > We sincerely thank for the thorough re-evaluation and the encouraging feedback. We are glad that the rebuttal has fully addressed your concerns.
> > >
> > > As suggested, we will integrate the architectural comparison table and the explicit explanation of the 25 Hz integer indexing mechanism into the camera-ready manuscript. Thank you again for the constructive dialogue that helped strengthen our paper.

---

### Official Review · Reviewer_95wX · 2026-03-03

**Soundness:** 3
**Presentation:** 2
**Significance:** 3
**Originality:** 3
**Overall Recommendation:** 5
**Confidence:** 5

**Summary:**

The paper proposes a method for sampling frequency independent (and decoupled) speech restoration (denoising and bandwidth extension). This is achieved via SFI-STFT front and backends, and applying additional query tokens along the frequency dimension at the bottleneck of the encoder-decoder structure. The encoder extracts "clean" features, which are used to condition the decoder via cross-attention with the query-extended features, before self-attention along the frequency dimension in the dual-path TF modelling setting.
The model is trained in two stages, first with perceptual and a log-scaled spectral reconstruction losses, followed by adversarial training with SFI-STFT discriminators and "human-feedback" losses.
Training is done exclusively on the VCTK dataset and evaluation across various scenarios, showing decent improvements over baselines.

**Compliance With Llm Reviewing Policy:**

Affirmed.

**Final Justification:**

Additional experiments and the promised modifications to the paper has addressed all my concerns and I will raise my score.

**Key Questions For Authors:**

See weaknesses above.

**Limitations:**

Yes.

**Strengths And Weaknesses:**

**Strengths:**
- The freq cross-self module along with extension queries is novel and shows benefits in bandwidth extension task.
- Ablation study covers and validates most of the architectural choices.

**Weaknesses:**
- The claim of the formulation of the de-coupled in/out sampling rates speech restoration might be abit overstated. The URGENT challenges already consider such a scenario, models must be SFI and includes bandwidth limitation on the input side. So, even though it doesn't explicitly use the de-coupled term, essentially it is a similar setting.
- The definition of assymetry is ambiguous, in the introduction, it is stated that the assymetry is in the size of encoder vs decoder, while in Sec. 3.3 the assymetry is described as the assymetry between the time and frequency paths in both the encoder and decoder.
- The motivation of using Mamba for the causal/streaming model is not clear, why not just apply causal mask in the time attention? What advantage does mamba specifically bring? This has not been ablated.
- The comparison of the URGENT 2025 blind test set is not fair, since the proposed model is trained exclusively on English full band VCTK data, it is natural that the UTMOS/NISQA is higher. Additionally, in the appendix, the results on the non-blind test set, where a dedicated model is trained and evaluated on the URGENT data, the performance is worse compared to other baselines, even the UTMOS and NISQA. Hence, in my opinion the advantage/disadvantage on the URGENT benchmark is not conclusive.
- The statement in Sec. 5.3 regarding the the proposed method not being specialized for perceptual enhancement is misleading, since the training losses include several perceptual components, including SSL feature based, as well as human-feedback loss terms.

---

> ### Author Rebuttal · Authors · 2026-03-27
>
> We sincerely thank the reviewer for the expert and detailed feedback. We address each point below.
>
> **W1: Decoupled formulation vs. URGENT.** We appreciate this clarification. While URGENT does consider bandwidth-limited inputs, the challenge still operates under a matched-rate SFI-STFT setting: inputs are assumed to be resampled to the target output rate before processing, and models operate on a shared spectral grid. This is precisely the conventional pipeline that our work aims to move beyond. For example, a 16kHz input at 48kHz output: URGENT resamples to 48kHz STFT with F=961 bins (only 321 informative, ~67% redundant); TF-Restormer processes F_E=321 natively, avoiding redundant computation. That said, we agree the "first decoupled formulation" claim is overstated. We will revise the framing to more precisely state our contribution: enabling direct processing of native input bandwidth to arbitrary output rates via separate SFI-STFT/iSTFT, without requiring external resampling.
>
> **W2: Ambiguity of "asymmetry."** We agree this terminology was ambiguous. In the revision, we will use "asymmetric encoder-decoder" specifically for the decoupled-rate role separation (observed-band analysis vs. unobserved-band synthesis), and avoid overloading for internal pathway differences.
>
> **W3: Motivation for Mamba in streaming mode.** The streaming model is intended as a feasibility demonstration rather than a core novelty claim. We do not claim Mamba outperforms causal attention in restoration quality—causal attention is a viable alternative and actually more intuitive option and may perform comparably. Our choice was purely deployment-oriented: for sustained streaming on long-form audio, Mamba's constant-memory inference avoids the linearly growing KV cache of causal attention.
>
> | | Causal Attention | Mamba (S6) |
> |---|---|---|
> | Time complexity (per layer) | O(T²d) | O(Tdn) |
> | Inference memory | O(Td), grows with T | O(dn), constant |
> | Long-form streaming | KV cache unbounded | Fixed-size state |
>
> where T = sequence length, d = hidden dim, n = SSM state dim (typically 16). This is consistent with Quan & Li (2024), who compared causal masked attention, retention, and Mamba for streaming speech enhancement and found Mamba to achieve the best overall performance while maintaining linear complexity. We acknowledge this was not ablated; we will include this rationale, the comparison, and the reference in the revision. The main contribution—the decoupled-rate asymmetric encoder-decoder with extension queries—is independent of the specific temporal module used in the streaming variant.
>
> *C. Quan and X. Li, "Multichannel Long-Term Streaming Neural Speech Enhancement for Static and Moving Speakers," in IEEE Signal Processing Letters, vol. 31, pp. 2295-2299, 2024
>
> **W4: Fairness of the URGENT comparison.** We agree that the URGENT blind-test result should be interpreted as OOD transfer evidence, not as a definitive controlled benchmark win. For Table 4(b), we reported the unified VCTK-trained model to evaluate generalization to unseen URGENT conditions. For the dedicated URGENT setting in the appendix, we also changed the loss balance to better align with the challenge objective, by reducing the generative/adversarial components and increasing regression-based fidelity losses. This shifts the model toward deterministic reconstruction, improving fidelity metrics but reducing perceptual metrics.
>
> We acknowledge that this loss rebalancing was not described clearly enough. To verify that this is the main cause of the perceptual drop, we trained a dedicated URGENT model with the original loss setting:
>
> | Setting | PESQ | ESTOI | SDR | MCD | LSD | sBERT | SpkSim | CAcc | UTMOS | NISQA | DNSMOS |
> |---|---|---|---|---|---|---|---|---|---|---|---|
> | original | 1.71 | 0.71 | 4.19 | 4.84 | 4.32 | 0.84 | 0.73 | 80.21 | 3.57 | 4.51 | 3.25 |
> | dedicated | 2.60 | 0.83 | 11.78 | 3.18 | 2.91 | 0.88 | 0.82 | 84.28 | 2.47 | 3.43 | 3.06 |
> | dedicated w/ original loss | 1.82 | 0.72 | 5.24 | 4.80 | 4.45 | 0.83 | 0.75 | 81.53 | 3.08 | 3.96 | 3.24 |
>
> The original-loss variant recovers perceptual metrics closer to the unified model, suggesting that the difference is primarily explained by the loss rebalancing rather than by the architecture itself. We also evaluated on DNS-Real and REVERB-Real (see our response to Reviewer cytr) as supplementary OOD evidence under genuinely mismatched real conditions, which helps contextualize the URGENT-specific results.
>
> **W5: "Not specialized for perceptual enhancement."** We agree this wording was misleading since our training includes SSL perceptual and human-feedback losses. We will revise.
>
> In summary, we will narrow the claim relative to URGENT, disambiguate "asymmetry," and present the URGENT results as transfer evidence with loss trade-off context.

---

> > ### Author Rebuttal · Reviewer_95wX · 2026-04-02
> >
> > Thanks for the detailed reply. Most of my concerns are addressed. However, please incorporate all the promised changes into the paper.
> >
> > Regarding the fairness of the URGENT comparison. It is paramount that the information about loss balancing should be included in the paper. Additionally, it is better to include at least one or two signal fidelity metrics in both Table 4(a) and 4(b). Since just these automatic MOS-like metrics does not always capture real signal fidelity. While it might not be impressive, it is well known that generative models will usually have lower signal fidelity scores, so I think it is ok. Or another intelligibility proxy metric like Character Accuracy or WER could also be provided to show that the results are both higher quality and inteligeble. The current tables with only the MOS-like scores might be a bit misleading as they show only one aspect.

---

> > > ### Author Response · Authors · 2026-04-06
> > >
> > > We appreciate the constructive feedback. We confirm that **all promised revisions** will be incorporated (contribution scope, "asymmetric" correction, streaming motivation, URGENT details). We agree that loss configuration critically determines the fidelity–quality trade-off, and will explain this alongside the URGENT results.
> > >
> > > ---
> > >
> > > Regarding additional metrics in Table 4, we fully agree that balanced evaluation covering both perceptual quality and signal/semantic fidelity should be a first priority—this is precisely our own position as well.
> > >
> > > However, the datasets in Table 4 lack reference signals or transcriptions, making intrusive fidelity metrics (e.g., PESQ, SI-SDR) and intelligibility proxies (e.g., WER) infeasible; no compared method reports them on these datasets either. We direct the reviewer to Appendix H (Tables 13–15), where we evaluate on the DNS 2020 simulated test set—entirely OOD—with full references available.
> > >
> > > Motivated by the reviewer's suggestion, we further conducted new experiments on the **REVERB** benchmark, also entirely OOD for TF-Restormer. In Tables A–B, WPE through DCN results are from prior works; discriminative baselines (TF-GridNet, MP-SENet, TF-Locoformer) were trained on REVERB data (and re-trained with DNS noise for REVERB-DNS). VoiceFixer and UNIVERSE++ are OOD restoration baselines. All results are averaged over far/near conditions.
> > >
> > > **Table A: REVERB-Sim (prior dereverberation methods)**
> > >
> > > |Model|CD↓|LLR↓|SNR_fw↑|PESQ↑|SRMR↑|
> > > |---|---|---|---|---|---|
> > > |Input|3.975|0.574|3.617|1.503|3.687|
> > > |WPE|3.748|0.514|4.864|1.722|4.220|
> > > |WRN|3.590|0.470|4.800|-|3.590|
> > > |GCRN|2.534|0.325|10.753|1.934|4.861|
> > > |TCN+SA|2.200|0.240|13.060|2.580|5.170|
> > > |D-MFMVDR|2.639|0.316|9.649|2.167|4.892|
> > > |DCN|**2.001**|**0.225**|**13.326**|**2.935**|5.269|
> > > |**TF-Restormer**|2.880|0.489|8.139|2.825|**8.004**|
> > >
> > > **Table B: REVERB-Sim (discriminative (in-domain) & restoration (OOD) baselines)**
> > >
> > > ||Fidelity|||||Perceptual|||||
> > > |---|---|---|---|---|---|---|---|---|---|---|
> > > |Model|CD↓|LLR↓|SNR_fw↑|PESQ↑|SRMR↑|UTMOS|DNSMOS|NISQA|SIG|BAK|
> > > |Input|3.975|0.574|3.617|1.503|3.687|1.455|1.365|1.255|1.750|1.390|
> > > |TF-GridNet|**2.110**|**0.216**|**14.458**|2.825|5.063|3.325|3.135|3.310|3.435|3.945|
> > > |MP-SENet|2.692|0.278|13.622|2.811|4.960|3.225|3.190|3.500|3.450|4.035|
> > > |TF-Locoformer|2.323|0.235|14.102|2.642|4.972|3.202|3.063|3.056|3.366|3.920|
> > > |VoiceFixer|3.708|0.946|7.992|1.278|4.938|2.858|3.088|4.072|3.388|3.927|
> > > |UNIVERSE++|5.213|0.901|5.731|1.220|5.124|2.370|2.835|4.145|3.150|3.835|
> > > |**TF-Restormer**|2.880|0.489|8.139|**2.825**|**8.004**|**4.130**|**3.335**|**4.860**|**3.585**|**4.115**|
> > >
> > > **Table C: REVERB-DNS test results**
> > >
> > > We constructed a REVERB-DNS set by mixing DNS noise at −10 to 0 dB SNR into REVERB-Sim data. TF-Restormer uses the same single model (no re-training); discriminative baselines were re-trained with DNS noise added.
> > >
> > > ||Fidelity|||||Perceptual|||||
> > > |---|---|---|---|---|---|---|---|---|---|---|
> > > |Model|CD↓|LLR↓|SNR_fw↑|PESQ↑|SRMR↑|UTMOS|DNSMOS|NISQA|SIG|BAK|
> > > |Input|5.694|0.962|-0.271|1.111|2.320|1.398|1.151|1.250|1.318|1.206|
> > > |TF-GridNet|3.654|**0.604**|**10.101**|1.867|5.696|2.587|2.961|2.905|3.285|3.824|
> > > |MP-SENet|4.347|0.780|7.807|1.493|5.217|2.072|2.641|2.607|3.082|3.416|
> > > |TF-Locoformer|4.118|0.630|8.262|1.581|5.388|2.159|2.517|2.537|2.921|3.402|
> > > |VoiceFixer|4.370|1.070|6.150|1.350|4.989|2.227|2.852|3.534|3.177|3.660|
> > > |UNIVERSE++|6.123|1.115|3.418|1.190|4.473|1.909|2.322|3.245|2.697|3.184|
> > > |**TF-Restormer**|**3.582**|0.707|7.130|**2.113**|**8.586**|**4.014**|**3.272**|**4.698**|**3.533**|**4.061**|
> > >
> > > On REVERB-Sim (Tables A–B), compared to other OOD restoration baselines (VoiceFixer, UNIVERSE++) which show severely degraded fidelity, TF-Restormer retains substantially stronger fidelity owing to its TF-domain discriminative backbone. In-domain discriminative models expectedly lead in CD/LLR/SNR_fw, though TF-Restormer's PESQ ties with TF-GridNet and SRMR far exceeds all baselines, leading all perceptual metrics by a large margin.
> > >
> > > On REVERB-DNS (Table C), where noise further increases the overall difficulty, TF-Restormer achieves the best CD and PESQ while dramatically outperforming all models in perceptual metrics—even against re-trained baselines. The fidelity gap observed on REVERB-Sim narrows or reverses here, demonstrating robustness as a general restoration model when severe reverberation and noise co-occur. We have added REVERB and REVERB-DNS samples to the demo page for qualitative comparison.
> > >
> > > We will add these results—along with DNS-Real and REVERB-Real (added in response to Reviewer cytr) and the newly conducted REVERB-Sim and REVERB-DNS evaluations—to the appendix, providing comprehensive fidelity analysis across diverse OOD conditions. Table 4 will clarify that its claims pertain to perceptual quality, directing readers to the appendix for balanced evaluation.

---

### Official Review · Reviewer_QLwr · 2026-03-11

**Soundness:** 3
**Presentation:** 3
**Significance:** 3
**Originality:** 3
**Overall Recommendation:** 4
**Confidence:** 4

**Summary:**

This paper proposes TF-Restormer, a speech restoration model designed for dynamic environments where the input and output sampling rates may differ. To address the mismatch between input and output bandwidths, the model adopts a query-based asymmetric encoder–decoder architecture. The encoder analyzes the input signal using a time–frequency dual-path structure to capture bandwidth-related characteristics, while the decoder synthesizes the enhanced signal using a frequency-extension query mechanism that enables flexible output bandwidth generation. The authors also introduce dedicated loss functions and a discriminator to improve training stability and restoration quality. Experimental results demonstrate competitive performance across several tasks, including speech restoration, speech enhancement, super-resolution, and real-recorded scenarios.

**Compliance With Llm Reviewing Policy:**

Affirmed.

**Final Justification:**

Based on the authors’ additional clarifications, I have revised my evaluation to “weak accept.” However, as the overall contribution remains somewhat limited, I am unable to assign a higher score.

**Key Questions For Authors:**

- In Table 6(b), the performance of w/F-proj. (sep.) and w/F-proj. (sha.) is compared. However, the main text does not clearly describe the difference between these two configurations. Could the authors clarify how these variants differ in terms of architecture or parameter sharing?
- Since some baseline methods appear to be trained under different conditions, it would be helpful if the authors could report the performance of TF-Locoformer under the same training setup used for TF-Restormer. Additionally, how does TF-Locoformer perform when scaled to a comparable model size (e.g., the 19M-parameter configuration used in TF-Restormer)?
- In Section 4.2, the paper mentions attaching multi-scale STFT discriminators, while the subsequent paragraph introduces the proposed multi-scale SFI-STFT discriminator. The relationship between these descriptions is not entirely clear. Could the authors clarify the discriminator architecture used during training and explain how it differs from the conventional multi-scale STFT discriminator?
- How does the proposed model perform when the input and output sampling rates are not observed during training? Additional discussion on the model’s generalization ability to unseen sampling-rate conditions, as well as the extent of performance degradation in such scenarios, would further strengthen the evaluation.

**Limitations:**

Please refer to W2 in the Strengths and Weaknesses section.

**Strengths And Weaknesses:**

$\textbf{S1. Flexible architecture for decoupled sampling-rate restoration.}$

The proposed query-based asymmetric architecture enables the model to handle input and output signals with different bandwidths, allowing restoration across varying sampling-rate conditions without explicit resampling procedures. This design improves flexibility for real-world applications where input bandwidth may vary.

$\textbf{S2. Training strategy for stable optimization.}$

The paper introduces several loss functions along with a unified discriminator to stabilize training and improve the perceptual quality of the restored speech.

$\textbf{S3. Competitive empirical performance across multiple tasks.}$

The proposed model demonstrates strong performance across multiple speech restoration tasks, including speech enhancement, super-resolution, and real-recorded scenarios, suggesting good practical applicability.

$\textbf{W1. Limited clarity regarding architectural novelty.}$

While the query-based asymmetric formulation is an interesting design choice for handling decoupled sampling rates, the overall architecture shares several design elements with existing time–frequency transformer-based speech restoration models. The paper would benefit from a clearer explanation and analysis of how the proposed design fundamentally differs from or improves upon prior approaches.

$\textbf{W2. Limited comparability of baselines.}$

Some baseline results appear to be taken from previously reported numbers that were trained under different datasets or training conditions. While this is common in the literature, additional comparisons under a unified training setup would make the evaluation more convincing.

$\textbf{W3. Imbalance between architectural description and experimental results.}$

The experimental section is extensive, but the description and analysis of the proposed architecture remain relatively limited. Providing more detailed motivation and analysis of the design choices would strengthen the technical contribution of the paper.

---

> ### Author Rebuttal · Authors · 2026-03-30
>
> We thank the reviewer for the detailed and constructive feedback.
>
> **W1: Difference from prior TF transformer models.** We appreciate this point and agree that the boundary between inherited and proposed components should be made clearer. We intentionally adopt an established TF dual-path backbone because it is well suited for SFI processing. Our contribution lies in the framework built on top of it for decoupled input-output rates: (1) encoding only the observed input band, (2) synthesizing unobserved output bands through extension queries, and (3) band-partitioned cross-attention so that missing-band queries attend only to observed-band encoder features. The novelty is a conditional information-flow structure tailored to decoupled-rate restoration. Crucially, when f_E=f_D, the MHCA path is bypassed entirely (enhancement-only mode); when f_E<f_D, cross-attention activates only for the unobserved extension region.
>
> | Aspect | TF-models (single-stage) | TF-Restormer Enc | TF-Restormer Dec |
> |---|---|---|---|
> | F-module | MHSA (or LSTM) | MHSA + **K/V proj**. | MHCA + MHSA + **K/V proj**. |
> | Freq. range | Resampled full F | **Observed F_E** | **Extended F_D** |
> | F-MHCA query | - | - | **Unobserved only (Q_ext)** |
> | F-MHCA K/V | - | - | Observed only from encoder |
>
> Table 5 confirms this is functionally important: enc-dec w/MHCA(small) (10.9M, 89G MACs) outperforms encoder-only (11.6M, 151G MACs) with fewer parameters and computation. We will distinguish adopted vs. proposed elements in the revision.
>
> **W2 / Q2: Baseline fairness and TF-Locoformer under the same setup.** We agree that fair comparison under matched training conditions is important. In Table 5, our encoder-only variant serves as the closest proxy to TF-Locoformer (both are symmetric TF dual-path models without the proposed asymmetric mechanism).
>
> We trained TF-Locoformer under our setup (7 blocks, parameter-matched). Symmetric full-band models become prohibitive at high output rates; we compare at the feasible 16kHz. Two conditions: (1) complex-domain L1 (original objective unstable), (2) matched loss:
>
> UNIVERSE (16kHz):
>
> | Model | PESQ | SDR | LSD | MCD | sBERT | UTMOS | DNSMOS |
> |---|---|---|---|---|---|---|---|
> | TF-Locoformer | 2.13 | 11.61 | 2.00 | 6.26 | 0.89 | 2.95 | 2.86 |
> | TF-Locoformer-our (1) | 2.06 | 10.32 | 1.91 | 6.07 | 0.88 | 3.12 | 2.91 |
> | TF-Locoformer-our (2) | 2.21 | 10.95 | 1.67 | 5.47 | 0.91 | 4.03 | 3.12 |
> | TF-Restormer | 2.30 | 11.12 | 1.45 | 5.08 | 0.91 | 4.08 | 3.25 |
>
> Under matched conditions (our(2)), TF-Restormer leads or matches across all metrics including SDR (11.12 vs 10.95). The original's higher SDR (11.61) reflects its regression-only training. VCTK+DEMAND shows the same pattern under matched loss (PESQ 3.41 vs 3.27, LSD 0.75 vs 0.89).
>
> In super-resolution, symmetric models cannot scale to high output rates; we report the feasible 8→16kHz alongside multi-rate results:
>
> VCTK SSR (noisy-distorted, LSD↓ / NISQA↑):
>
> | Model | 8→16 | 8→44.1 | 16→48 |
> |---|---|---|---|
> | TF-Locoformer-our | 2.00 / 3.92 | — | — |
> | encoder-only | 2.23 / 3.72 | 2.31 / 3.80 | 2.24 / 3.92 |
> | TF-Restormer(small) | 1.48 / 4.20 | 1.56 / 4.21 | 1.60 / 4.23 |
> | TF-Restormer | **1.16 / 4.49** | **1.18 / 4.52** | **1.18 / 4.54** |
>
> This scalability limitation of symmetric models ("—") is precisely a core motivation for our asymmetric design.
>
> **W3: Architecture description and analysis.** We appreciate this suggestion. The core rationale is the functional asymmetry between analysis and synthesis: the encoder requires deeper processing for robust features under diverse distortions, while the decoder can be lightweight because it synthesizes missing bands conditioned on encoder features. This is validated by Table 5 and Appendix D.2 (Figure 5), which illustrates three decoder variants. We will expand this in the revision with information-flow diagrams and capacity scaling analysis.
>
> **Q1: F-proj shared vs. separate.** In the shared configuration, a single projection matrix A_h is used across all frequency modules (both encoder and decoder) and all key–value mappings. In the separate configuration, each module has its own independent matrix. We will clarify this in the revision.
>
> **Q3: Discriminator clarification.** The multi-scale STFT discriminator (Défossez et al., 2023) is our starting point. Our modification replaces the fixed-rate STFT with SFI-STFT (constant frame duration across rates). The Conv2D architecture remains the same; only the front-end is changed to be rate-agnostic. Table 6(a) confirms this shared SFI design outperforms separate rate-specific discriminators. We will clarify this in the revision.
>
> **Q4: Unseen sampling rates.** Appendix E (Table 11) shows stable performance when each rate appears with ≥10% training frequency, with degradation only at extreme imbalance (1%). For completely unseen rates, uninitialized query regions would degrade performance; incorporating structural priors is a promising future direction.

---

> > ### Author Rebuttal · Reviewer_QLwr · 2026-04-02
> >
> > Thank you for the detailed explanations and for providing additional experimental results. All of my concerns have been adequately addresses.

---

> > > ### Author Response · Authors · 2026-04-02
> > >
> > > We sincerely thank for the positive re-evaluation. We will incorporate the suggested clarifications—including the F-proj shared/separate distinction, the SFI-STFT discriminator description, and expanded architectural analysis—into the camera-ready version. Thank you for the constructive feedback throughout the review process.

---

### Official Review · Reviewer_cytr · 2026-03-15

**Soundness:** 3
**Presentation:** 3
**Significance:** 3
**Originality:** 3
**Overall Recommendation:** 4
**Confidence:** 3

**Summary:**

This paper studies speech restoration when the input sampling rate and desired output sampling rate are decoupled, rather than matched as in most prior restoration and super-resolution systems. The paper proposes TF-Restormer, a query-based asymmetric encoder–decoder in the complex STFT domain: a heavier TF dual-path encoder analyzes only the observed input bandwidth, while a lightweight decoder reconstructs missing high-frequency content using learnable extension queries and frequency cross-attention. The method is paired with a shared SFI-STFT discriminator for multi-rate adversarial training and a scaled log-spectral loss intended to stabilize optimization under severe distortions. The authors evaluate the approach on universal restoration, denoising, super-resolution, real-recorded data, and a streaming variant, and report generally strong trade-offs between signal fidelity and perceptual quality across heterogeneous input-output rate settings.

**Compliance With Llm Reviewing Policy:**

Affirmed.

**Key Questions For Authors:**

How robust is the unified model to input-output rate pairs or degradation combinations that are rare or absent during training? A more explicit out-of-distribution analysis would strengthen the significance of the “single model across arbitrary rates” claim.

The paper notes that the VCTK noisy-distorted super-resolution evaluation may slightly favor the proposed method because the training simulation is similar. Can the authors quantify this concern, for example with an evaluation on a more mismatched real or public benchmark?

Can the authors provide more detail on the practical streaming setup, including chunking, buffering, and whether the reported 80 ms latency includes all algorithmic lookahead and projection effects? A clearer explanation would strengthen the real-time claim.

Do the authors have any human listening test results, even on a small subset, to validate the heavy reliance on non-intrusive perceptual predictors? A positive answer would increase my confidence in the perceptual claims

**Limitations:**

yes

**Strengths And Weaknesses:**

strengths
 The core idea—decoupling input-band analysis from output-band synthesis via an asymmetric TF encoder–decoder with extension queries—is well aligned with the problem being posed. The formulation of decoupled input–output rates is also meaningful: rather than forcing everything onto a shared target grid via external resampling, the model directly analyzes the native observed bandwidth and predicts an STFT at the requested output rate. That framing is one of the paper’s most compelling contributions.

A major strength is that the architecture is not just “another restoration backbone,” but is explicitly designed around the multi-rate setting. The encoder-only analysis of the observed band and decoder-side extension queries are sensible and well motivated by the figures and text. The frequency cross-self module appears especially important, and the ablation shows that the encoder–decoder design with MHCA is both more effective and often more efficient than encoder-only alternatives, particularly for larger output rates. The authors proceed to analyze a general aspect of restoration system design here: how to separate observed-band inference from unobserved-band synthesis in a way that scales better than symmetric full-band processing. That makes the work more broadly interesting than a narrowly tuned benchmark submission.

The empirical coverage is diverse which includes UNIVERSE-style general restoration, VCTK+DEMAND denoising, VCTK super-resolution across multiple rate conversions, real-recorded VoxCeleb samples, and the URGENT 2025 blind test set. The results tables also reflect a thoughtful effort to position the model relative to vocoder-based, diffusion-based, TF-based, and task-specific baselines. The offline model appears especially competitive in balancing fidelity and perceptual quality, while the streaming variant remains reasonably strong under causal constraints.

The ablations are useful and materially strengthen the paper. In particular, the comparisons on the decoder design, shared SFI discriminator, frequency projection module, and scaled log-spectral loss help support the main modeling choices rather than leaving them as unverified design decisions. The appendix snippet on runtime is also helpful: the paper is not just claiming flexibility, but providing model size, MACs, RTF, and latency for the proposed variants, including an 80 ms streaming latency.


Weaknesses


First, while the paper is technically solid, some of the novelty claims are stated a bit too strong. The paper claims “the first decoupled formulation,” but the core ingredients—SFI-STFT, TF dual-path processing, asymmetric encoder–decoder design, and query-based latent completion—are all connected to prior ideas. I do think the combination is meaningful and well executed, but the novelty is better framed as a well-motivated synthesis for multi-rate speech restoration than as a wholly new conceptual direction. The paper would benefit from a more careful positioning of what is strictly new versus what is adapted and recombined.

Although the evaluation breadth is a strength, the evidence for the paper’s strongest robustness claims still comes primarily from synthetic or semi-synthetic settings. The inclusion of VoxCeleb and URGENT results is valuable, but much of the main controlled comparison remains based on VCTK-derived simulations and the UNIVERSE benchmark. This means the paper demonstrates strong performance under standardized evaluation pipelines, yet the case for generalization to genuinely mismatched real-world degradations is still somewhat underdeveloped. The concern is reinforced by the authors’ own observation that the VCTK noisy-distorted super-resolution setup follows a training-like simulation procedure and may therefore slightly favor the proposed method. I appreciate that this limitation is acknowledged, but it suggests that the paper would be substantially stronger with additional evaluation on public datasets that better stress real acoustic mismatch. In particular, CHiME-5 or CHiME-6 would test robustness on real far-field conversational speech recorded in everyday environments; DNS Challenge real recordings would provide a stronger assessment under real noisy conditions; and REVERB or WHAMR! would help evaluate performance under reverberation and more realistic distortion patterns. As written, the empirical results are promising, but the evidence for broad real-world robustness remains somewhat narrower than the headline framing suggests

While the scaled log-spectral loss is well motivated intuitively and supported by ablations, the rationale for the particular adaptive scaling rule still feels somewhat heuristic. The paper would be stronger with a clearer discussion of why this objective should be preferred, in theory or in practice, over other robust losses widely used in audio restoration, such as L1, Huber, or Charbonnier losses, as well as established spectral objectives like log-magnitude STFT or multi-resolution STFT losses, or scale-invariant waveform-level criteria such as SI-SDR. At present, the empirical gains are encouraging, but the conceptual case for this specific design choice is not yet fully developed.

The paper leans heavily on non-intrusive MOS proxies such as UTMOS, DNSMOS, and NISQA in several places. These are useful, but they are still predictors, not human listening tests. For a paper making perceptual quality claims across diverse restoration settings, a small controlled human study would have made the conclusions much stronger, especially on the real-recorded or blind-test portions. Overall, a notable area considered by the study is the trade-off between perceptual naturalness and signal fidelity across restoration regimes, but the perceptual side would be more convincing with at least limited human validation.

---

> ### Author Rebuttal · Authors · 2026-03-27
>
> We sincerely thank the reviewer for the careful and constructive review. We address each point below.
>
> **W1: Novelty framing.** We agree that our original wording was too strong. Our contribution is not a new primitive operator, but a new conditional composition for decoupled input-output rates: observed-band-only encoding, extension-query-based decoding for unobserved bands, and rate-consistent multi-rate training. We will revise the paper accordingly and avoid overstating the claim as a wholly new conceptual direction.
>
> **W2 / Q2: Real-world robustness and possible VCTK bias.** The possible training-like bias applies only to the VCTK noisy-distorted super-resolution setting in Table 3, not to the rest of the evaluation. We note that the paper already includes real-data evaluation in Table 4; to further strengthen this aspect, we additionally evaluated on DNS Challenge 2020 real recordings and the REVERB Challenge. These datasets better match our single-speaker restoration scope than CHiME-5/6 (multi-speaker) or WHAMR! (separation-focused).
>
> DNS-Real results:
>
> | Model | SIG | BAK | OVRL | UTMOS | NISQA |
> |---|---|---|---|---|---|
> | Input | 2.985 | 2.510 | 2.212 | 1.940 | 2.160 |
> | Miipher | 3.325 | 3.976 | **3.171** | **3.911** | 4.124 |
> | VoiceFixer | 3.174 | 3.919 | 2.875 | 2.351 | 3.529 |
> | Resemble-Enhance | 3.395 | 3.993 | 3.100 | 2.767 | 4.329 |
> | UNIVERSE++ | 2.999 | 3.660 | 2.641 | 2.306 | 3.317 |
> | AnyEnhance | **3.488** | 3.977 | 3.161 | - | - |
> | TF-Restormer | 3.399 | **4.047** | 3.127 | 3.502 | **4.360** |
>
> REVERB-Real results:
>
> | Model | SIG | BAK | OVRL | UTMOS | NISQA | SRMR |
> |---|---|---|---|---|---|---|
> | Input | 1.676 | 1.481 | 1.347 | 1.558 | 1.716 | 3.180 |
> | TF-GridNet | 3.298 | 3.758 | 2.933 | 2.342 | 1.954 | **7.319** |
> | MP-SENet | 3.308 | 3.934 | 3.014 | 2.094 | 1.840 | 7.007 |
> | TF-Locoformer | 3.232 | 3.706 | 2.847 | 2.129 | 1.642 | 6.639 |
> | TF-Restormer | **3.557** | **4.101** | **3.303** | **3.140** | **2.127** | 7.156 |
>
> These results show competitive performance under genuinely mismatched real conditions. We also note that the VCTK noisy-distorted test set uses held-out speakers, separate noise/RIR sources, and distinct distortion ranges, so there is no direct data overlap with training.
>
> **W3: Scaled log-spectral loss rationale.** We appreciate the request for broader comparison. Appendix D.1 includes a gradient plot (Figure 4), and we will expand this discussion in the revision. The gradient of our loss is ∂ℓ/∂d = w/(d+w), where d = |y−s| and w_{tf} = E[|S_{m,t,f}|]. Unlike ℓ₁ (constant gradient) or ℓ₂ (gradient ∝ d), our formulation yields gradients strongest near d ≈ 0 and naturally attenuating for large d, prioritizing refinement in well-aligned spectral regions.
>
> Waveform SI-SDR is less suitable under compound distortions. Compared with Huber and Charbonnier, which limit sensitivity through clipped/saturated gradients, our loss more explicitly prioritizes small-error refinement. Table 7 compares these variants; the adaptive version outperforms all alternatives including fixed-w log1p (w = 1).
>
> Regarding the adaptive weighting, we conducted additional analysis and found that spectral errors are strongly heteroscedastic:
>
> | Dataset | Utt. ρ | Freq. ρ | CV raw→norm | Reduction |
> |---|---|---|---|---|
> | VCTK-DEMAND | 0.49 | 0.96 | 2.36→1.35 | 43% |
> | VCTK-SR(ND) 8→16k | 0.50 | 0.94 | 1.62→0.42 | 74% |
> | VCTK-SR(ND) 8→44.1k | 0.64 | 0.99 | 2.28→0.34 | 85% |
> | VCTK-SR(ND) 16→48k | 0.58 | 0.98 | 2.65→0.28 | 89% |
>
> Freq. ρ = per-frequency-bin Spearman correlation between E[|S(f)|] and E[|S(f)−Ŝ(f)|] (all p ≈ 0); Utt. ρ = per-utterance correlation. The per-bin correlation is near-perfect (0.94–0.99), and normalizing by source magnitude reduces the coefficient of variation by up to 89%. This supports that the adaptive weighting addresses the heteroscedastic structure of spectral errors, consistent with variance-stabilization principles. We will include this analysis in the revision.
>
> **W4 / Q4: Human listening tests.** We agree that non-intrusive MOS predictors are not substitutes for human listening. We do not have a formal listening study within the rebuttal timeline; accordingly, we restrict our perceptual claims to consistent improvements across multiple independent predictors rather than human-validated conclusions. We provide an anonymous demo page as qualitative support.
>
> **Q1: OOD robustness.** Appendix E, Table 11 reports this analysis. Performance remains stable when each rate appears with at least 10% training frequency; degradation appears only at extreme imbalance (1%). Within the trained span, extension queries on a shared physical-frequency grid enable stable generalization.
>
> **Q3: Streaming setup.** 40ms window, 20ms hop, per-STFT-frame. Non-causal Conv2D adds two frames of lookahead. Total: 40ms + 2×20ms = 80ms, inclusive of all lookahead.
>
> In summary, we will narrow the novelty claim, expand the loss analysis, and have provided DNS-Real/REVERB results above.

---

### Decision · Program_Chairs · 2026-04-30

**Decision:**

Accept (regular)

**Comment:**

TF-Restormer proposes a query-based asymmetric framework for speech restoration at decoupled input-output rates, using frequency extension queries and cross-attention for spectral reconstruction. Scores range from 3–5 (avg. 4.0). The rebuttal was generally effective, with most reviewers acknowledging their concerns were addressed, though one outlier score remains.

The paper's core strength is its problem framing: reviewers appreciated that it is explicitly designed around the multi-rate restoration setting rather than being another generic backbone (cytr, QLwr), and the frequency cross-attention with extension queries was recognized as a genuinely novel module with demonstrated benefit (95wX). Empirical coverage across multiple tasks and datasets was also noted positively (cytr, QLwr, PB8a).

The most consistent weakness across reviewers is limited architectural novelty in isolation — while the overall formulation is coherent, the individual components connect closely to prior work, and several reviewers (cytr, QLwr, PB8a) felt the novelty claims were overstated. 95wX raises a related but distinct point: the paper uses two conflicting definitions of "asymmetry" (encoder/decoder size vs. time/frequency path structure), which should be resolved.

Evaluation methodology is another recurring concern. cytr notes the over-reliance on non-intrusive objective metrics, and no listening tests were added despite this being flagged. The URGENT 2025 benchmark comparison is contested by 95wX on fairness grounds — the proposed model's advantage on the blind test set is confounded by training data mismatch, and on the non-blind set it underperforms baselines, making the benchmark results inconclusive. cytr also notes that robustness claims rest largely on synthetic data, though the authors did provide some real-world evaluation (DNS Challenge, REVERB-Real) in the rebuttal. QLwr's baseline comparability concern was cleanly resolved by retraining under matched conditions.

PB8a remains the only reviewer at weak reject, despite the authors addressing their concerns — no score adjustment was made, which should be noted.

Recommendation: Accept. The formulation is meaningful, the rebuttal addressed the majority of concerns substantively, and the cross-attention extension query mechanism is a solid contribution. The authors should reconcile the asymmetry definition inconsistency, temper novelty claims appropriately, and are strongly encouraged to include at least a small-scale listening test in the final version.